# Understanding the Detrimental Class-level Effects of Data Augmentation

**Polina Kirichenko**[1,2]   **Mark Ibrahim**[2]   **Randall Balestriero**[2]   **Diane Bouchacourt**[2]
**Ramakrishna Vedantam**[2]   **Hamed Firooz**[2]   **Andrew Gordon Wilson**[1]

[1]New York University      [2]Meta AI

## Abstract

Data augmentation (DA) encodes invariance and provides implicit regularization critical to a model's performance in image classification tasks. However, while DA improves average accuracy, recent studies have shown that its impact can be highly class dependent: achieving optimal average accuracy comes at the cost of significantly hurting individual class accuracy by as much as $20\%$ on ImageNet. There has been little progress in resolving class-level accuracy drops due to a limited understanding of these effects. In this work, we present a framework for understanding how DA interacts with class-level learning dynamics. Using higher-quality multi-label annotations on ImageNet, we systematically categorize the affected classes and find that the majority are inherently ambiguous, co-occur, or involve fine-grained distinctions, while DA controls the model's bias towards one of the closely related classes. While many of the previously reported performance drops are explained by multi-label annotations, our analysis of class confusions reveals other sources of accuracy degradation. We show that simple class-conditional augmentation strategies informed by our framework improve performance on the negatively affected classes.

## 1   Introduction

Data augmentation (DA) provides numerous benefits for training of deep neural networks including promoting invariance and providing regularization. In particular, DA significantly improves the generalization performance in image classification problems when measured by average accuracy [24, 19, 2, 18]. However, Balestriero et al. [1] and Bouchacourt et al. [9] showed that strong DA, in particular, Random Resized Crop (RRC) used in training of most modern computer vision models, may disproportionately hurt accuracies on some classes, e.g. with up to $20\%$ class-level degradation on ImageNet compared to milder augmentation settings (see Figure 1 left). Performance degradation even on a small set of classes might result in poor generalization on downstream tasks related to the affected classes [55], while in other applications it would be unethical to sacrifice accuracy on some classes for improvements in average accuracy [28, 7, 63, 10].

Balestriero et al. [1] attempted to address class-level performance degradation by applying DA selectively to the classes where the accuracy improves with DA strength. Surprisingly, they found that this augmentation policy did not address the issue and the performance on non-augmented classes still degraded with augmentation strength. In this work we perform detailed analysis and explore the mechanisms causing the class-level performance degradation. In particular, we identify the *interactions between class-conditional data distributions* as the cause of the class-level performance degradation with augmentation: DA creates an overlap between the data distributions associated with different classes. As a simple illustrative example, in Figure 1 (right) we show that the standard RRC operation creates an overlap between the "car" and "wheel" classes. As a result, the model learns to predict label "car" on "wheel" images, and the performance on the "wheel" class drops. Importantly,

37th Conference on Neural Information Processing Systems (NeurIPS 2023).

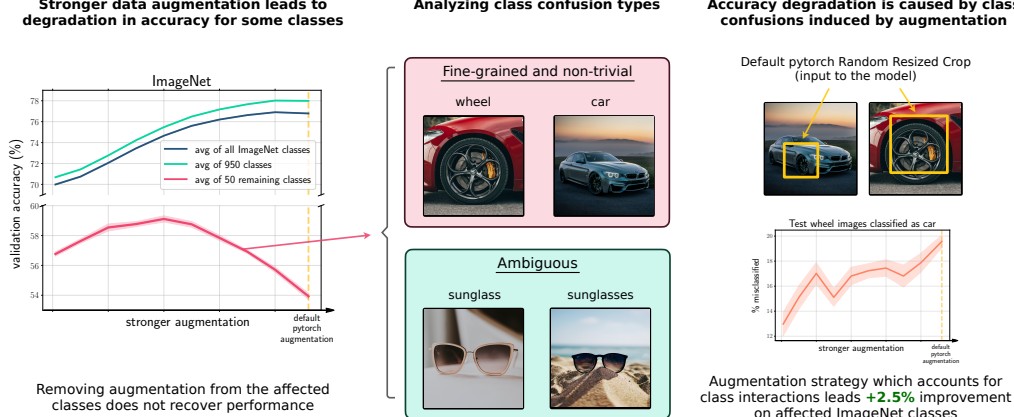

Figure 1: **We show that the classes negatively affected by data augmentation are often ambiguous, co-occurring or fine-grained categories and analyze how data augmentation exacerbates class confusions.** **Left:** Average accuracy of ResNet-50 on ImageNet against Random Resized Crop (RRC) data augmentation strength: average of all classes (blue), average of the 50 classes on which stronger RRC hurts accuracy the most (red), and the average of the remaining 950 classes (green). Yellow line indicates the default RRC setting used in training of most computer vision models. **Middle:** We systematically categorize the types of class confusions exacerbated by strong data augmentation: while some of them include ambiguous or correlated classes, there is a number of fine-grained and non-trivial confusions. **Right:** Often the class-level accuracy drops due to overlap with other classes after applying augmentation: e.g. heavily augmented samples from "car" class can look like typical images from "wheel" class. As a result, the model learns to predict "car" on "wheel" images, and the accuracy on the "wheel" class drops. To resolve the negative effect of strong augmentation on classes like "wheel", we should modify augmentation strength of classes like "car".

if we want to improve the performance on the "wheel" class, we need to modify the augmentation policy on the class "car" and not "wheel" as was done in prior work [1]. We summarize our findings in Figure 1. In particular, our contributions are the following:

- We refine the analysis of class-level effects of data augmentations by correcting for label ambiguity. Specifically, we use multi-label annotations on ImageNet [4] and measure the effects of data augmentation for each class in terms of the original and multi-label accuracy. Through this analysis, we find that class-level performance degradation reported in Balestriero et al. [2] and Bouchacourt et al. [9] is overestimated (Section 4).

- We systematically categorize the class confusions exacerbated by strong augmentation and find that many affected classes are ambiguous or co-occurring and are often affected by label noise (Figure 1 middle and Section 5). We focus on addressing the remaining fine-grained and non-trivial class confusions.

- We show that for addressing DA biases it is important to consider the classes with an increasing number of *false positive mistakes*, and not only the classes negatively affected in accuracy. By taking into account our observations on DA affecting class interactions, we propose a simple class-conditional data augmentation strategy that leads to improvement on the affected group of classes by $2.5\%$ on ImageNet (Section 6). This improvement is in contrast to the previously explored class-conditional DA in Balestriero et al. [1] which failed to improve class-level accuracy.

- We confirm our findings across multiple computer vision architectures including ResNet-50 [22], EfficientNet [61] and ViT [15] (Section F), multiple data augmentation transformations including Random Resized Crop, mixup [73], RandAugment [13] and colojitter (Section G), as well as two additional datasets besides ImageNet (Section G).

## 2 Related work

**Understanding data augmentation, invariance and regularization.** Hernández-García and König [24] analyzed the DA from the perspective of implicit regularization. Botev et al. [8] propose an explicit regularizer that encourages invariance and show that it leads to improved generalization. Balestriero et al. [2] derive an explicit regularizer to simulate DA to quantify its benefits and limitations and estimate the number of samples for learning invariance. Gontijo-Lopes et al. [19] and Geiping et al. [18] study the mechanisms behind the effectiveness of DA, which include data diversity, exchange rates between real and augmented data, additional stochasticity and distribution shift. Bouchacourt et al. [9] measure the learned invariances using DA. Lin et al. [37] studied how data augmentation induces implicit spectral regularization which improves generalization.

**Biases of data augmentations.** While DA is commonly applied to improve generalization and robustness, a number of prior works identified its potential negative effects. Hermann et al. [23] showed that decreasing minimum crop size in Random Resized Crops leads to increased texture bias. Shah et al. [56] showed that using standard DA amplifies model's reliance on spurious features compared to models trained without augmentations. Idrissi et al. [30] provided a thorough analysis on how the strength of DA for different transformations has a disparate effect on subgroups of data corresponding to different factors of variation. Kapoor et al. [33] suggested that DA can cause models to misinterpret uncertainty. Izmailov et al. [31] showed that DA can hurt the quality of learned features on some classification tasks with spurious correlations. Balestriero et al. [1] and Bouchacourt et al. [9] showed that strong DA may disproportionately hurt accuracies on some classes on ImageNet, and in this work we focus on understanding this class-level performance degradation through the lens of interactions between classes.

**Multi-label annotations on ImageNet.** A number of prior works identified that ImageNet dataset contains label noise such as ambiguous classes, multi-object images and mislabeled examples [4, 57, 69, 46, 59, 45]. Tsipras et al. [67] found that nearly 20% of ImageNet validation set images contain objects from multiple classes. Hooker et al. [27] ran a human study and showed that examples most affected by pruning a neural network are often mislabeled, multi-object or fine-grained. Yun et al. [72] generate pixel-level multi-label annotations for ImageNet train set using a large-scale computer vision model. Beyer et al. [4] provide re-assessed (ReaL) multi-label annotations for ImageNet validation set which aim to resolve label noise issues, and we use ReaL labels in our analysis to refine the understanding of per-class effects of DA.

**Adaptive and learnable data augmentation.** Xu et al. [70] showed that data augmentation may exacerbate data bias which may lead to model' suboptimal performance on the original data distribution. They propose to train the model on a mix of augmented and unaugmented samples and then fine-tune it on unaugmented data after training which showed improved performance on CIFAR dataset. Raghunathan et al. [49] showed standard error in linear regression could increase when training with original data and data augmentation, even when data augmentation is label-preserving. Rey-Area et al. [51] and Ratner et al. [50] learn DA transformation using GAN framework, while Hu and Li [29] study the bias of GAN-learned data augmentation. Fujii et al. [17] take into account the distances between classes to adapt mixed-sample DA. Hauberg et al. [21] learn class-specific DA on MNIST. Numerous works, e.g. Cubuk et al. [12], Lim et al. [36], Ho et al. [25], Hataya et al. [20], Li et al. [35], Cubuk et al. [13], Tang et al. [62], Müller and Hutter [43] and Zheng et al. [74] find dataset-dependent augmentation strategies. Benton et al. [3] proposed Augerino framework to learn augmentation form training data. Zhou et al. [75], Cheung and Yeung [11], Mahan et al. [40] and Miao et al. [41] learn class- or input-dependent augmentation policies. Yao et al. [71] propose to modify mixed-sample augmentation to improve out-of-domain generalization.

**Robustness and model evaluation beyond average accuracy.** While Miller et al. [42] showed that model's average accuracy is strongly correlated with its out-of-distribution performance, there is a number of works that showed that only evaluating average performance can be deceptive. Teney et al. [64] showed counter-examples for "accuracy-on-the-line" phenomenon. Kaplun et al. [32] show that while model's average accuracy improves during training, it may decrease on a subset of examples. Sagawa et al. [54] show that training with Empirical Risk Minimization may lead to suboptimal performance in the worst case. Bitterwolf et al. [6] evaluated ImageNet models' performance in terms of a number of metrics beyond average accuracy, including worst-class accuracy and precision. Richards et al. [52] demonstrate that improved average accuracy on ImageNet and standard out-of-distribution ImageNet variants may lead to exacerbated geographical disparities.

# 3 Evaluation setup and notation

Since we aim to understand class-level accuracy degradation emerging with strong data augmentation reported in Balestriero et al. [1], we closely follow their experimental setup. We focus on ResNet-50 models [22] trained on ImageNet [53] and study how average and class-level performance changes depending on the Random Resized Crop augmentation strength: it is by far the most widely adopted augmentation which leads to significant average accuracy improvements and is used for training the state-of-the-art computer vision models [58, 65, 38]. We train ResNet-50 for $88$ epochs using label smoothing with $\alpha = 0.1$ [60]. We use image resolution $R_{train} = 176$ during training and evaluate on images with resolution $R_{test} = 224$ following Balestriero et al. [1], Touvron et al. [66] and `torchvision` training recipe[1]. More implementation details can be found in Appendix A.

**Data augmentation.** We apply random horizontal flips and Random Resized Crop (RRC) DA when training our models. In particular, for an input image of size $h \times w$ the RRC transformation samples the crop scale $s \sim U[s_{low}, s_{up}]$ and the aspect ratio $r \sim U[r_{low}, r_{up}]$, where $U[a, b]$ denotes a uniform distribution between $a$ and $b$. RRC then takes a random crop of size $\sqrt{shwr} \times \sqrt{shw/r}$ and resizes it to a chosen resolution $R \times R$. We use the standard values for $s_{up} = 100\%$ and aspect ratios $r_{low} = 3/4, r_{up} = 4/3$, and vary the lower bound of the crop scale $s_{low}$ (for simplicity, we will further use $s$) between $8\%$ and $100\%$ which controls *the strength of augmentation*: $s = 8\%$ corresponds to the strongest augmentation (note this is the default value in `pytorch` [48] RRC implementation) and $s = 100\%$ corresponds no cropping hence no augmentation. For each augmentation strength, we train 10 models with different random seeds.

**ReaL labels.** Beyer et al. [4] used large-scale vision models to generate new label proposals for ImageNet validation set which were then evaluated by human annotators. These Reassessed Labels (ReaL) correct the label noise present in the original labels including mislabeled examples, multi-object images and ambiguous classes. Since there are possibly multiple ReaL labels for each image, model's prediction is considered correct if it matches one of the plausible labels. Further we will use $l_{ReaL}(x)$ to denote the set of ReaL labels of example $x$.

**Evaluation metrics.** We aim to measure performance of model $f_s(x)$ as a function of DA strength, i.e. RRC lower bound of the crop scale $s$. We measure average accuracy $a(s)$, and per-class accuracy $a_k(s)$ with respect to both original ImageNet labels and ReaL multi-label annotations given by:

$$a_k^{or}(s) = 1/|X_k| \sum_{x \in X_k} I[f_s(x) = k] \quad \text{and} \quad a_k^{ReaL}(s) = 1/|X_k| \sum_{x \in X_k} I[f_s(x) \in l_{ReaL}(x)],$$

where $X_k$ are images from class $k$ in validation set. We will refer to $a^{or}$ and $a^{ReaL}$ as *original accuracy* and *ReaL accuracy*, respectively. To quantify the class accuracy drops, Balestriero et al. [1] compare the per-class accuracy of models trained with the strongest DA ($s = 8\%$) and models trained without augmentation ($s = 100\%$ which effectively just resizes input images without cropping), while Bouchacourt et al. [9] compared class-level accuracy of models trained with RRC with $s = 8\%$ and models trained with fixed size Center Crop. In our analysis, we evaluate per-class accuracy drops comparing the maximum accuracy attained on a particular class $k$ across all augmentation levels $\max_s a_k(s)$ and accuracy on that class when training with strongest DA $a_k(s = 8\%)$. We will refer to the classes with the highest accuracy degradation computed by:

$$\Delta a_k = \max_s a_k(s) - a_k(s = 8\%)$$

as the classes *most negatively affected* by DA (with respect to either original labels or multi-label ReaL annotations). To summarize performance on the affected classes, we will evaluate average accuracy of classes with the highest $\Delta a_k$ (in many cases focusing on $5\%$ of ImageNet classes with the highest accuracy drop following Balestriero et al. [1]). In the following sections, we also highlight the importance of measuring other metrics beyond average and per-class accuracy which comprise a more thorough evaluation of DA biases.

While we focus on the analysis of ResNet-50 on ImageNet with RRC augmentation as the main setup following prior work [1, 9], we additionally confirm our observations on other architectures (Efficient-Net [61] and ViT [14]) in Appendix F, other data augmentation transformations (RandAugment [13], colorjitter and mixup [73]), and other datasets (CIFAR-100 and Flowers102 [44]) in Appendix G.

---

[1] https://pytorch.org/blog/how-to-train-state-of-the-art-models-using-torchvision-latest-primitives/

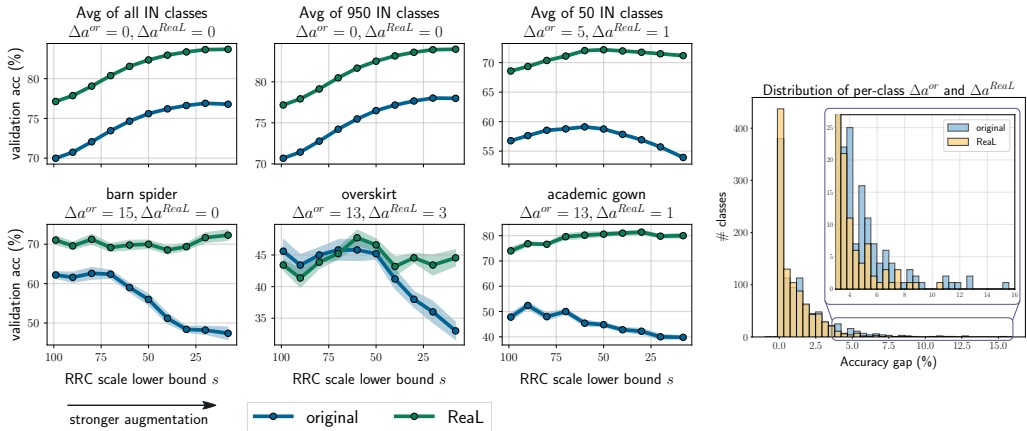

Figure 2: **We find that for many classes the negative effects of strong data augmentation are muted if we use high-quality multi-label annotations. Left**: Average and per-class accuracy of ResNet-50 trained on ImageNet evaluated with original and ReaL labels as a function of Random Resized Crop augmentation strength ($s = 8\%$ corresponds to the strongest and default augmentation). The top row shows the average accuracy of all ImageNet classes, the 50 classes with the highest original accuracy degradation and the remaining 950 classes. The bottom row shows the accuracy of 3 individual classes most significantly affected in original accuracy when using strong augmentation. **Right**: Distribution of per-class accuracy drops $\Delta a_k$ for original and ReaL labels. The distribution of $\Delta a_k^{or}$ has a heavier tail compared to the one computed with ReaL labels.

## 4 Per-class accuracy degradation with strong data augmentation is overestimated due to label ambiguity

Previous studies reported that the performance of ImageNet models is effectively better when evaluated using re-assessed multi-label annotations which address label noise issues in ImageNet [4, 57, 69]. These works showed that recent performance improvements on ImageNet might be saturating, but the effects of this label noise on granular per-class performance has not been previously studied. In particular, it is unclear how correcting for label ambiguity would affect the results of Balestriero et al. [1] and Bouchacourt et al. [9] on the effects of DA on class-level performance.

We observe that **for many classes with severe drops in accuracy on original labels, the class-level ReaL multi-label accuracy is considerably less affected.** The right panel of Figure 2 shows the distributions of per-class accuracy drops $\Delta a_k^{or}$ and $\Delta a_k^{ReaL}$, and we note that the distribution of $\Delta a_k^{or}$ has a much heavier tail. Using multi-label accuracy in evaluation reveals there are much fewer classes which have severe effective performance drop: e.g. only 37 classes with $\Delta a_k^{ReaL} > 4\%$ as opposed to 83 classes with $\Delta a_k^{or} > 4\%$, moreover, there are no classes with $\Delta a_k^{ReaL} > 11\%$.

On the left panel of Figure 2, we show how multi-label accuracy evaluation impacts the average and individual class performance across different augmentation strengths $s$. In particular, in the top row plots we see that while the average accuracy of all classes follows a similar trend when evaluated with either original or ReaL labels, the average accuracy of 50 classes most negatively affected in original accuracy only decreases by $1\%$ with ReaL labels as opposed to more significant $5\%$ drop with original labels. The bottom row shows the accuracy for "barn spider", "overskirt" and "academic gown" classes which have the highest $\Delta a_k^{or}$, and accuracy trends for all 50 most negatively affected classes are shown in Appendix C. For many of these classes which are hurt in original accuracy by using stronger DA, the ReaL accuracy is much less affected. For example, for the class "barn spider" the original accuracy is decreased from $63\%$ to $47\%$ if we use the model trained with RRC $s = 8\%$ compared to $s = 70\%$, while the highest ReaL accuracy is achieved by the strongest augmentation setting on this class.

However, there are still classes for which the ReaL accuracy degrades with stronger augmentation, and in Appendix C we show ReaL accuracy trends against augmentation strength $s$ for 50 classes with the highest $\Delta a^{ReaL}$. While some of them (especially classes from the "animal" categories)

may still be affected by the remaining label noise [69, 68, 57, 39, 4], for other classes the strongest DA leads to suboptimal accuracy. In the next section, we aim to understand why strong DA hurts the performance on these classes. We analyze model's predictions and consistent confusions on these classes and find that the degradation in performance is caused by the interactions between class-conditional distributions induced by DA.

# 5   Data augmentation most significantly affects classification of ambiguous, co-occurring and fine-grained categories

In this section, we aim to understand the reasons behind per-class accuracy degradation when using stronger data augmentation by analyzing the most common mistakes the models make on the affected classes and how they evolve as we vary the data augmentation strength. We consider the classes most affected by strong DA (see Figures in Appendix C) which do not belong to the "animal" subtree category in the WordNet hierarchy [16] since fine-grained animal classes were reported to have higher label noise in previous studies [68, 57, 39, 4]. We focus on the 50 classes with the highest $\Delta a_k^{or}$ (corresponding to $\Delta a_k^{or} > 5\%$), and 50 classes with the highest $\Delta a_k^{ReaL}$ (corresponding to $\Delta a_k^{ReaL} > 4\%$). For a pair of classes $k$ and $l$ we define the confusion rate (CR) as:

$$CR_{k \to l}(s) = 1/|X_k| \sum_{x \in X_k} I[f_s(x) = l],$$

i.e. the ratio of examples from class $k$ misclassified as $l$. For each affected class, we identify most common confusions and track the CR against the RRC crop scale lower bound $s$. We also analyze the reverse confusion rate $CR_{l \to k}(s)$.

We observe that in many cases DA strength controls the model's preference in predicting one or another plausible ReaL label, or preference among semantically similar classes. We roughly outline the most common types of confusions on the classes which are significantly affected by DA. The different types of confusion differ in the extent to which the accuracy degradation can be attributed to label noise versus the presence of DA. We also characterize how DA effectively changes the data distribution of these classes leading to changes in performance. These categories are closely related to common mistake types on ImageNet identified by Beyer et al. [4] and Vasudevan et al. [69], but we focus on class-level interactions as opposed to instance-level mistakes and particularly connect them to the impact of DA. We use *semantic similarity* and *ReaL labels co-occurence* as a criteria to identify a confusion category for a pair of classes. We can measure semantic similarity by (a) WordNet class similarity, given by the Wu-Palmer score which relies on the most specific common ancestor of the class pair in the WordNet tree, and (b) similarity of the class name embeddings[2]. To estimate the intrinsic distribution overlap of the classes, we compute ReaL labels co-occurrence between classes $k$ and $l$ as:

$$C_{kl} = \sum_{x \in X} I[k \in l_{ReaL}(x)] \times I[l \in l_{ReaL}(x)] / \sum_{x \in X} I[k \in l_{ReaL}(x)],$$

which is the ratio of examples that have both labels $k$ and $l$ among the examples with the label $k$. Using these metrics, depending on a higher or lower semantic similarity and higher or lower ReaL labels overlap, we categorize confused class pairs as *ambiguous*, *co-occurring*, *fine-grained* or *semantically unrelated*. Below we discuss each category in detail, and the examples are shown in Figure 3 and Appendix Figure 6. We provide more details on computing the metrics for identifying the confusion type and categorize the confusions of all affected classes in Appendix D.

**Class-conditional distributions induced by DA.**    To aid our understanding of the class-specific effects of DA, it is helpful to reason about how a parametrized class of DA transformations $\mathcal{T}_s(\cdot)$ changes the distributions of each class in the training data $p_k(x)$. We denote the augmented class distributions by $\mathcal{T}_s(p_k)$. In particular, if supports of the distributions $\mathcal{T}_s(p_k)$ and $\mathcal{T}_s(p_l)$ for two classes $k$ and $l$ overlap, the model is trained to predict different labels $k$ and $l$ on similar inputs corresponding to features from both classes $k$ and $l$ which will lead to performance degradation. Some class distributions $p_k$ and $p_l$ are intrinsically almost coinciding or highly overlapping in the ImageNet dataset, while others have distinct supports, but in all cases the parameters of DA $s$ will control the overlap of the induced class distributions $\mathcal{T}_s(p_k)$ and $\mathcal{T}_s(p_l)$, and thus the biases of the model when making predictions on such classes.

---

[2]We use `NLTK` library [5] for WordNet and `spaCy` library [26] for embeddings similarity.

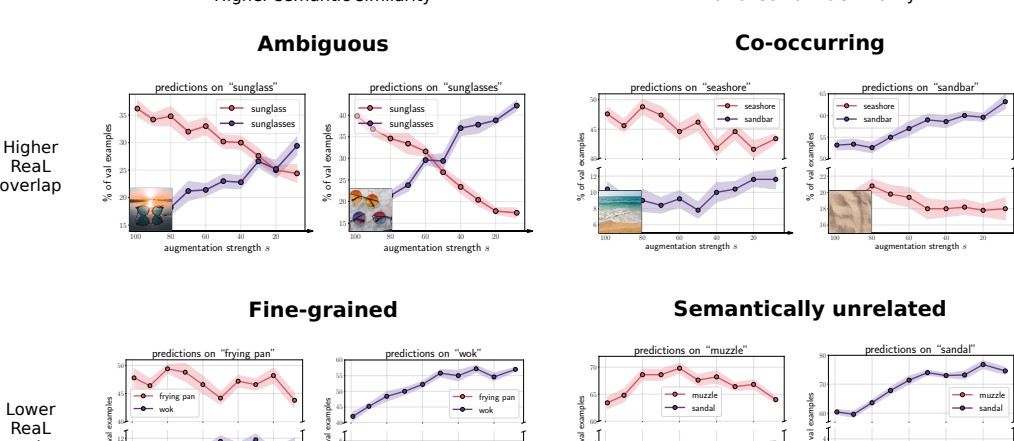

Figure 3: **Types of class confusions affected by data augmentation with varied semantic similarity and data distribution overlap.** Each panel shows a pair of confused classes which we categorize into: *ambiguous*, *co-occurring*, *fine-grained* and *semantically unrelated*, depending on the inherent class overlap and semantic similarity. For each confused class pair, the left subplot corresponds to the class $k$ whose accuracy decreases with strong data augmentation (DA), e.g. "sunglass" on top left panel: the ratio of validation samples from that class which are classified correctly decreases with stronger DA, while the confusion rate with another class $l$ (e.g. class "sunglasses" on top left panel) increases. The right subplot shows the percent of examples from class $l$ that get classified as $k$ or $l$ against DA strength.

**Intrinsically ambiguous or semantically (almost) identical classes.** Prior works [e.g. 4, 57, 69, 67] identified that some pairs of ImageNet classes are practically indistinguishable, e.g. "sunglasses" and "sunglass", "monitor" and "screen", "maillot" and "maillot, tank suit". These pairs of classes generally have higher semantic similarity and higher ReaL labels overlap $C_{kl}$. We observe that in many cases the accuracy on one class within the ambiguous pair degrades with stronger augmentations, while the accuracy on another class improves. The supports of distributions of these class pairs $p_k$ and $p_l$ highly overlap or even coincide, but with varying $s$ depending on how the supports of $\mathcal{T}_s(p_k)$ and $\mathcal{T}_s(p_l)$ overlap the model would be biased towards predicting one of the classes. In Figure 3 top left panel, we show how the frequencies of most commonly predicted labels change on an ambiguous pair of classes "sunglass" and "sunglasses" as we vary the crop scale parameter. The ReaL labels for these classes overlap with $C_{kl} = 87\%$, and the majority of confusions are resolved after accounting for multi-label annotations. We note that for images from both classes the frequency of "sunglasses" label increases with stronger DA while "sunglass" predictions have the opposite trend. Models trained on ImageNet often achieve a better-than-random-guess accuracy when classifying between these classes due to overfitting to marginal statistical differences and idiosyncrasies of the data labeling pipelines. While DA strength controls model's bias towards predicting one or another plausible label, the models are not effectively making mistakes when confusing such classes.

For the remaining categories described below, the class distributions become more overlapping when strong DA is applied during training, and data augmentation amplifies or causes problematic misclassification.

**Co-occurring or overlapping classes.** There is a number of classes in ImageNet which correspond to semantically different objects which often appear together, e.g. "academic gown" and "mortarboard", "Windsor tie" and "suit", "assault rifle" and "military uniform", "seashore" and "sandbar". These pairs of classes have rather high overlap in ReaL labels depending on the spurious correlation strength, and their semantic similarity can vary, but generally it would be lower than for ambiguous classes. The class distributions of co-occurring classes inherently overlap, however, stronger DA may increase the overlap in class distribution supports. For example, with RRC we may augment the

sample such that only the spuriously co-occurring object, but not the main object, is left in the image, and the model would still be trained to predict the original label: e.g. we can crop just the mortarboard in an image labeled as "academic gown". It was previously shown that RRC can increase model's reliance on spurious correlations [23, 56] which can lead to meaningful mistakes, not explained by label ambiguity. In Figure 3 top right panel, we show how DA strength impacts model's bias towards predicting "sandbar" or "seashore" class (these classes co-occur with $C_{kl} = 64\%$ and the majority of confusions are resolved by accounting for ReaL labels). We emphasize that unlike the ambiguous classes, the co-occuring classes cause meaningful mistakes on the test data, which are not always resolved by multi-label annotations. For example, the model will be biased to predict "academic gown" even when shown an image of just the mortarboard.

**Fine-grained categories.** There is a number of semantically related class pairs like "tobacco shop" and "barbershop", "frying pan" and "wok", "violin" and "cello", where objects appear in related contexts, share some visually similar features and generally represent fine-grained categories of a similar object type. These classes have high semantic similarity and do not have a significant ReaL label overlap (sometimes such examples are affected by mislabeling but such classes generally do not co-occur). The class distributions for such categories are close to each other or slightly overlapping, but strong DA pulls them closer, and $\mathcal{T}(p_i)$ and $\mathcal{T}(p_k)$ would be more overlapping due to e.g. RRC resulting in the crops of the visually similar features or shared contexts in the augmented images from different categories. In Figure 3 bottom left panel, we show how model's confusion rates change depending on RRC crop scale for fine-grained classes "frying pan" and "wok" (these classes rarely overlap with only $C_{kl} = 9\%$).

**Semantically unrelated.** In the rare but most problematic cases, the stronger DA will result in confusion of semantically unrelated classes. While they share similar low-level visual features, they are semantically different, their distributions $p_k$ and $p_l$ and ReaL labels do not overlap, and they get confused with one another specifically because of strong DA), for example, categories like "muzzle" and "sandal", "bath towel" and "pillow". Figure 3 bottom right panel shows how confusions between unrelated classes "bath towel" and "pillow" emerge with stronger DA.

In Appendix D we show a larger selection of example pairs from each confusion category. Among the confusions on the most significantly affected classes, approximately $53\%$ are fine-grained, $21\%$ are co-occurring, $16\%$ are ambiguous and the remaining $10\%$ are semantically unrelated. While the confusion of semantically unrelated categories is the most rare, it is potentially most concerning since it corresponds to more severe mistakes.

While in some cases we can intuitively explain why the model becomes biased to predict one of the two closely related classes with stronger DA (such as with "car" and "wheel" classes in Figure 1), in other cases the bias arises due to statistical differences in the training data such as data imbalance. ImageNet train set is not perfectly balanced with 104 classes out of 1000 containing less training examples than the remaining classes. We observed that $21\%$ of these less represented classes were among the ones most significantly affected in original or ReaL per-class accuracy. Since DA can push class-conditional distributions of related classes closer together, if one of such classes is less represented in training data it may be more likely to be impacted by stronger augmentation. In Appendix C we show how average accuracy on the underrepresented classes changes depending on the data augmentation strength.

We observe that using other data augmentation transformations such as RandAugment [13] or colorjitter on ImageNet, or mixup [73] on CIFAR-100 results in similar effects of exacerbated confusions of related categories (see Appendix section G). In particular, RandAugment and colorjitter often affect class pairs which are distinct in their color or texture, while mixup on CIFAR-100 may amplify confusions of classes within the same superclass.

## 6 Class-conditional data augmentation policy interventions

In Section 5, we showed that class-level performance degradation occurs with stronger augmentation because of the interactions between different classes. The models tend to consistently misclassify images of one class to another related, e.g. co-occurring, class. In this section, we use these insights to develop a class-conditional augmentation policy and improve the performance on the classes negatively affected by DA.

Table 1: **Class-conditional data augmentation policy informed by our insights improves performance on the negatively affected classes while maintaining high overall accuracy.** Average accuracy of different augmentation policies on all ImageNet classes, negatively affected classes and remaining majority of classes.

| Augmentation strategy | | Avg acc | Avg acc of 50 classes | Avg acc of 950 classes |
|---|---|---|---|---|
| Standard DA | $s = 8\%$ | $76.79_{\pm 0.03}$ | $53.93_{\pm 0.20}$ | $77.99_{\pm 0.02}$ |
| | $s = 60\%$ | $74.65_{\pm 0.03}$ | $59.11_{\pm 0.20}$ | $75.47_{\pm 0.02}$ |
| Class-cond. Balestriero et al. [1] | | $76.11_{\pm 0.05}$ | $43.02_{\pm 0.28}$ | $77.85_{\pm 0.04}$ |
| Our class-cond. DA | $m = 10$ | $76.70_{\pm 0.03}$ | $54.99_{\pm 0.15}$ | $77.84_{\pm 0.03}$ |
| | $m = 30$ | $76.70_{\pm 0.03}$ | $55.48_{\pm 0.23}$ | $77.82_{\pm 0.03}$ |
| | $m = 50$ | $76.68_{\pm 0.04}$ | $56.34_{\pm 0.14}$ | $77.75_{\pm 0.04}$ |

Balestriero et al. [1] identified that DA leads to degradation in accuracy for some classes and also showed that a naive class-conditional augmentation approach is not sufficient for removing these negative effects. In particular, using oracle knowledge of validation accuracies of models pretrained with different augmentation levels, they evaluated a DA strategy where augmentation is applied to all classes except the ones with degraded accuracy (i.e. classes with $\Delta a_k^{or} > 0$) which are instead processed with Center Crop. Since this approach didn't recover the accuracy on the affected classes, they hypothesized that DA induces a general invariance or an implicit bias that still negatively affects classes that are not augmented when DA is applied to the majority of the training data.

We explore a simple class-conditional augmentation strategy based on our insights regarding class confusions. By changing the augmentation strength for as few as $1\%$ to $5\%$ of classes, we observe substantial improvements on the classes negatively affected by the standard DA. We also provide an alternative explanation for why the class-conditional augmentation approach from Balestriero et al. [1] was not effective. In particular, we found in many cases that as we train models with stronger augmentation, a class $k$ negatively affected by DA consistently gets misclassified as a related class $l$ (see Section 5). We can precisely describe these confusions in terms of *False Negative* (FN) mistakes for class $k$ (not recognizing an instance from class $k$) and *False Positive* (FP) mistakes for class $l$ (misclassifying an instance from another class as class $l$):

$$FN_k^{or}(s) = \sum_{x \in X_k} I[f_s(x) \neq k] \quad \text{and} \quad FP_l^{or}(s) = \sum_{(x \in X) \cap (x \notin X_l)} I[f_s(x) = l].$$

We argue that to address the degraded accuracy of class $k$ it is also important to consider DA effect on class $l$. In Appendix E, we show the number of class-level False Positive mistakes as a function of DA strength for the set of classes with the highest $\Delta FP_l = FP_l(s = 8\%) - \min_s FP_l(s)$ (i.e. the classes for which the number of FP mistakes increased the most with standard augmentation). Note that these classes are often semantically related to and confused with the ones affected in accuracy, for example "stage" is confused with "guitar", "barbershop" with "tobacco shop".

We explore a simple adaptation of DA policy informed by the following observations: (1) generally stronger DA is helpful for the majority of classes and leads to learning more diverse features, (2) a substantially increased number of False Positive mistakes for a particular class likely indicates that its augmented data distribution overlaps with other classes and it might negatively affect accuracy of those related classes. Further, we discuss training the model from scratch using class-conditional augmentation policy, and in Appendix E we consider fine-tuning the model using class-conditional augmentation starting from a checkpoint pre-trained with the strongest augmentation strength.

**Class-conditional augmentation policy.** In general, for a class that is not closely related to other classes, strong RRC augmentation should lead to learning diverse features correlated with the class label and improving accuracy. Thus, by default we set the strongest data augmentation value $s = 8\%$ for the majority of classes, and change augmentation level for a small subset of classes. We change the augmentation strength for $m$ classes for which False Positive mistakes grew the most with stronger DA (i.e. the classes with the highest $\Delta FP_l$). However, completely removing augmentation from these classes would hurt their accuracy (or equivalently increase the number

of False Negative mistakes) so we need to balance the tradeoff between learning diverse features and avoiding class confusions. As a heuristic, we set augmentation strength for each class $l$ to $s^* = \arg\min_s FP_l(s) + FN_l(s)$ corresponding to the minimum of the total number of class-related mistakes across augmentation levels.

We vary the number of classes for which we change augmentations in the range $m \in \{10, 30, 50\}$. In Table 1 we show the results where the parameters of the augmentation policy are defined using original ImageNet labels and in Appendix E we use ReaL multi-label annotations. We compare this intervention to the baseline model trained with the strongest augmentation $s = 8\%$, mild augmentation level $s = 60\%$ optimal for average accuracy on the affected set of classes, and the class-conditional augmentation approach studied in Balestriero et al. [1] where we remove augmentation from the negatively affected classes. We find existing approaches sacrifice accuracy on the subset of negatively affected classes for overall average accuracy or vice versa. For example, as we previously observed the default model trained with $s = 8\%$ achieves high average accuracy on the majority of classes but suboptimal accuracy on the 50 classes affected by strong augmentation. Optimal performance on these 50 classes is attained by the model trained with $s = 60\%$, but the overall average accuracy significantly degrades. Removing augmentation from the negatively affected classes only exacerbates the effect and decreases the accuracy both on the affected set and on average. At the same time, tuning down augmentation level on 1 to $5\%$ of classes with the highest FP mistake increase improves the accuracy on the affected classes by $2.5\%$ for $m = 50$, and taking into account the tradeoff between False Positive and False Negative mistakes helps to maintain high average accuracy overall and on majority of classes (the average accuracy decreased by $0.1\%$ for $m = 50$). These results support our hypothesis and demonstrate how a simple intervention on a small number of classes informed by the appropriate metrics can substantially improve performance on the negatively affected data.

In Appendix F we study the class-level accuracy degradation of the ViT-S [15] model trained on ImageNet with varied RRC augmentation strength. We observe that a similar set of classes is negatively affected in accuracy with stronger augmentation and multi-label annotations resolve some cases of accuracy degradation. Similarly, we identify the same categories of class confusions. By conducting class-conditional augmentation intervention and adapting augmentation strength for $m = 10$ classes with the highest increase in False Positive mistakes, we improve the average accuracy on the degraded classes by over $3\%$: from $52.28\% \pm 0.18\%$ to $55.49\% \pm 0.07\%$.

## 7 Discussion

In this work, we provide new insights into the class-level accuracy degradation on ImageNet when using standard data augmentation. We show that it is important to consider the interactions among class-conditional data distributions, and how data augmentation affects these interactions. We systematically categorize the most significantly affected classes as inherently ambiguous, co-occurring, or involving fine-grained distinctions. These categories often suffer from label noise and thus the overall negative effect is significantly muted when evaluating performance with cleaner multi-label annotations. However, we also identify non-trivial cases, where the negative effects of data augmentation cannot be explained by label noise such as fine-grained categories. Finally, in contrast to prior attempts, we show that a simple class-conditional data augmentation policy based on our insights can significantly improve performance on the classes negatively affected by standard data augmentation.

**Practical recommendations.** When evaluating model performance, one should not only check average accuracy, which may conceal class-level learning dynamics. Instead, we recommend researchers also consider other metrics such as False Negative and False Positive mistakes to better detect which confusions data augmentation introduces or exacerbates. In particular, when training a model with strong augmentations, one should train another model with milder augmentation to check whether these finer-grained metrics degraded as an indicator that augmentation is biasing the model's predictions. We can then design targeted augmentation policies to improve performance on the groups negatively affected by standard augmentation.

## Acknowledgements

We thank Yucen Lily Li, Sanae Lotfi, Pavel Izmailov, David Lopez-Paz and Badr Idrissi for useful discussions. Polina Kirichenko and Andrew Gordon Wilson were partially supported by NSF CAREER IIS-2145492, NSF I-DISRE 193471, NIH R01DA048764-01A1, NSF IIS-1910266, NSF 1922658 NRT-HDR, Meta Core Data Science, Google AI Research, BigHat Biosciences, Capital One, and an Amazon Research Award.

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

## Appendix Outline

This appendix is organized as follows:

- In Section A, we provide additional details on the training procedure and hyper-parameters used in our experiments.
- In Section B we provide precise definitions of our evaluation metrics.
- In Section C we visualize the accuracy on the classes most negatively affected by data augmentation.
- We discuss the different types of class confusions caused by data augmentation in Section D.
- In Section E we provide additional experiments for the proposed class-conditional data augmentation policy.
- In Section F we provide additional results with EfficientNet and Vision Transformer architectures.
- In Section G we confirm our results on additional data augmentation transformations and two addditional datasets.
- We discuss broader impacts and limitations in Section H.

## A    Training details

Following [1], we train ResNet-50 models for 88 epochs with SGD with momentum $0.9$, using batch size $1024$, weight decay $10^{-4}$, and label smoothing $0.1$. We use cyclic learning rate schedule starting from the initial learning rate $10^{-4}$ with the peak value $1$ after $2$ epochs and linearly decaying to $0$ until the end of training. We use PyTorch [47], automatic mixed precision training with `torch.amp` package[3], `ffcv` package [34] for fast data loading. We use image resolution $176$ during training, and resolution $224$ during evaluation, following Touvron et al. [66] and `torchvision` training recipe[4]. Balestriero et al. [1] also use different image resolution at training and test time: ramping up resolution from $160$ to $192$ during training and evaluating models on images with resolution $256$. We train $10$ independent models with different random seeds for each augmentation strength $s \in \{8, 20, 30, 40, 50, 60, 70, 80, 90, 99\%\}$ where $s = 8\%$ corresponds to the strongest and default augmentation.

## B    Evaluation metrics

To understand the biases introduced or exacerbated by data augmentation, we use a number of fine-grained metrics and evaluate them for models trained with different augmentation levels. We compute these metrics using original ImageNet validation labels and ReaL multi-label annotations [4]. We use $f_s(\cdot)$ to denote a neural network trained with augmentation parameter $s$, $l_{ReaL}(x)$ a set of ReaL labels for a validation example $x$, $X$ a set of all validation images, $X_k$ the validation examples with the original label $k$.

**Accuracy.** The average accuracy for original and ReaL labels is defined as:

$$a^{or}(s) = 1/|X| \sum_{x \in X} I[f_s(x) = k] \quad \text{and} \quad a^{ReaL} = 1/|X| \sum_{x \in X} I[f_s(x) \in l_{ReaL}(x)],$$

while for per-class accuracies $a_k^{or}(s)$ and $a_k^{ReaL}(s)$ the summation is over the set $X_k$ instead of all validation examples $X$. The accuracy on class $k$ with original labels $a_k^{or}(s)$ also correspond to the *recall* of the model on that class.

**Confusion.** In Section 5 we looked at class confusions, in particular, for a pair of classes $k$ and $l$ the confusion rate (CR) is defined as:

---

[3] https://pytorch.org/docs/stable/amp.html
[4] https://pytorch.org/blog/how-to-train-state-of-the-art-models-using-torchvision-latest-primitives/

$$CR_{k \to l}(s) = 1/|X_k| \sum_{x \in X_k} I[f_s(x) = l],$$

i.e. the ratio of examples from class $k$ misclassified as $l$. We are only discussing confusions $CR_{k \to l}$ in the context of original labels.

**False Positive and False Negative mistakes.** In Section 6, we emphasized the importance of looking at how data augmentation impacts not only per-class accuracy but also the number of *False Positive* (FP) mistakes for a particular class:

$$FP_k^{or}(s) = \sum_{(x \in X) \cap (x \notin X_k)} I[f_s(x) = k] \quad \text{and} \quad FP_k^{ReaL}(s) = \sum_{(x \in X) \cap (k \notin l_{ReaL}(x))} I[f_s(x) = k]$$

for original and Real labels respectively. The number of *False Negative* mistakes on class $k$ in terms of the original labels are directly related to the accuracy, or recall, on that class:

$$FN_k^{or}(s) = \sum_{x \in X_k} I[f_s(x) \neq k] = |X_k|(1 - a^{or}(s)),$$

while for multi-label annotations we define it as:

$$FN_k^{ReaL}(s) = \sum_{(x \in X) \cap (k \in l_{ReaL}(x))} I[f_s(x) \notin l_{ReaL}(x)],$$

i.e. the number of examples $x$ which were misclassidied by the model where $k$ was in the ReaL label set $l_{ReaL}(x)$. In Section 6 we explored $s_k^* = \arg \min FN_k(s) + FN_k(s)$ as a proxy for optimal class-conditional augmentation level which emphasizes the inherent tradeoff between class-level accuracy and the number of False Positive mistakes.

**Affected classes.** We are focusing on analyzing model's behavior on the classes which were negatively affected by strong (default) augmentation in terms of original or ReaL accuracy, i.e. classes where the accuracy drop $\Delta a_k = a_k(s_k^*) - a_k(s = 8\%)$ from $a_k(s_k^*) = \max_s a_k(s)$ to $a_k(s = 8\%)$ is the highest. We focus on $5\%$ of classes (50 classes) with the highest $\Delta a_k$ following Balestriero et al. [1] and measure the average accuracy on this set of classes as a function of $s$ and after interventions in Section 6. In Section 6, we also look at classes where the number of FP mistakes increased the most with strong DA, i.e. with the highest $\Delta FP_k = FP_k(s = 8\%) - \min_s FP_k(s)$.

## C   Accuracy of the classes most negatively affected by data augmentation

We show the per-class accuracies as a function of data augmentation strength $s$ for (1) the 50 classes most negatively affected in original accuracy, i.e. with the highest $\Delta a_k^{or}$ in Figure 4, and (2) 50 classes most negatively affected in ReaL accuracy in Figure 5.

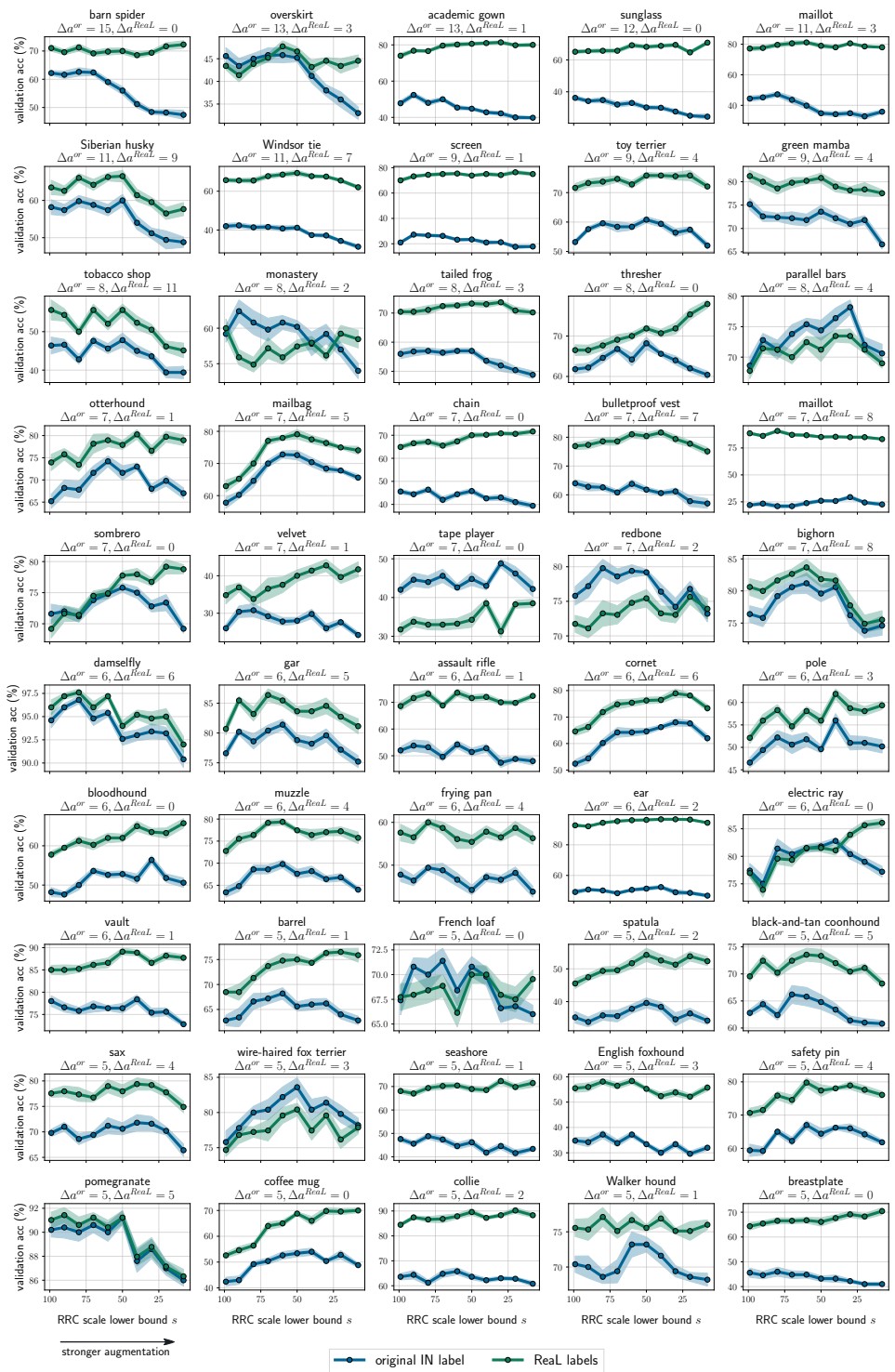

Figure 4: Per-class class validation accuracies of ResNet-50 trained on ImageNet computed with original and ReaL labels as a function of Random Resized Crop data augmentation scale lower bound $s$. We show the accuracy trends for the classes with the highest difference between the maximum accuracy on that class across augmentation levels $\max_s a_k^{or}(s)$ and the accuracy of the model trained with $s = 8\%$. On each subplot below the name of the class we show the accuracy drops with respect to original and ReaL labels: $\Delta a_k^{or}$ and $\Delta a_k^{ReaL}$. We report the mean and standard error over 10 independent runs of the network.

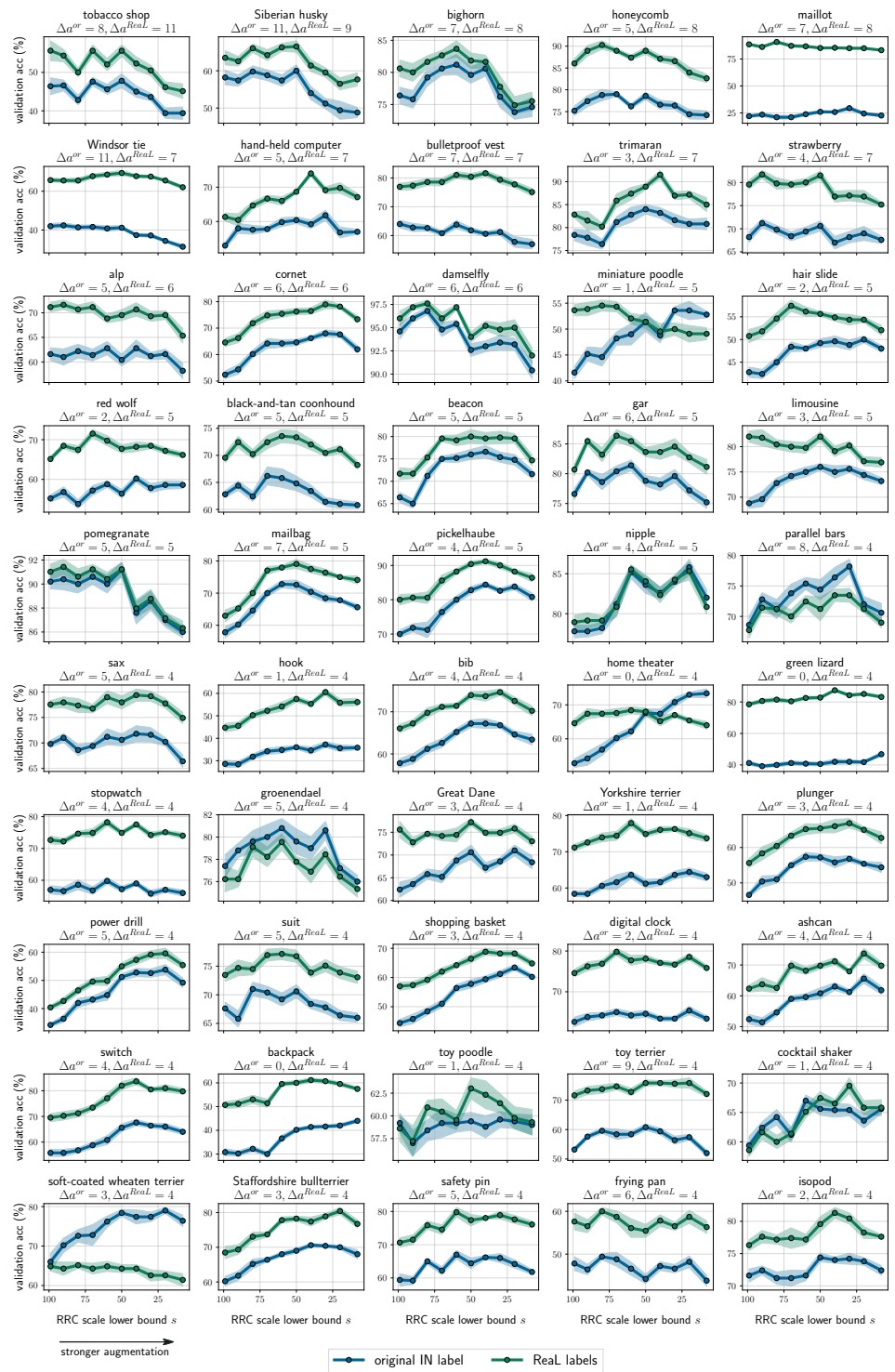

Figure 5: Per-class class validation accuracies of ResNet-50 trained on ImageNet computed with original and ReaL labels as a function of Random Resized Crop data augmentation scale lower bound $s$. We show the accuracy trends for the classes with the highest difference between the maximum ReaL accuracy on that class across augmentation levels $\max_s a_k^{ReaL}(s)$ and the ReaL accuracy of the model trained with $s = 8\%$. On each subplot below the name of the class we show the accuracy drops with respect to original and ReaL labels: $\Delta a_k^{or}$ and $\Delta a_k^{ReaL}$. We report the mean and standard error over 10 independent runs of the network.

# D Class confusion types

In Table 2 we show the classes most negatively affected in accuracy by strong data augmentation (column "Affected class $k$") and the confusions the model starts making more frequently with stronger augmentation ("Confused class $l$"). In particular, we study the union of 50 classes most affected in original accuracy and 50 classes most affected in ReaL accuracy (see Section C) which do not belong to the animal subtree in WordNet tree. We focus on the confusions $l$ where confusion rate difference

$$\Delta CR_{k \to l} = CR_{k \to l}(s = 8\%) - \min_s CR_{k \to l}(s)$$

is the highest for class $k$ and above 2.5% (see Section B for definition of confusion rate $CR_{k \to l}(s)$). Additionally for each pair of confused classes $k$ and $l$ we also look at

$$\Delta CR^*_{l \to k} = \max_s CR_{l \to k}(s) - CR_{l \to k}(s = 8\%)$$

which characterizes to what extent the model trained with weaker augmentation starts making the reverse confusion more often compared to the strong DA model.

To quantitatively estimate the confusion type for each pair of classes, we measure the intrinsic distribution overlap of the classes and their semantic similarity. We compute the overlap in Real labels for classes $k$ and $l$, which is the ratio of examples that have both labels $k$ and $l$ among the examples with the label $k$:

$$C_{kl} = \sum_{x \in X} I[k \in l_{ReaL}(x)] \times I[l \in l_{ReaL}(x)] / \sum_{x \in X} I[k \in l_{ReaL}(x)]$$

and intersection-over-union of the two classes:

$$IoU_{kl} = \sum_{x \in X} I[k \in l_{ReaL}(x)] \times I[l \in l_{ReaL}(x)] / \sum_{x \in X} I[k \in l_{ReaL}(x) \text{ or } l \in l_{ReaL}(x)].$$

We use WordNet class similarity and similarity of word embeddings from spaCy [26] to measure semantic similarity. Note that these metrics only serve as approximate measures of distribution overlap and semantic distance since (1) the ReaL labels still contain some amount of label noise and may contain mislabelled examples or examples that are missing some of the plausible labels, (2) the WordNet distance is sometimes low for classes that are semantically very similar, and (3) spaCy doesn't have a representation for all words and is underestimating the similarity of closely related concepts. However, all together these metrics can point towards one of the appropriate confusion type categories.

In Figure 6 we show more examples of the confusion rates for different pairs of classes $k$ and $l$ as a function of data augmentation strength $s$ where $k$ is among the ones most negatively affected in accuracy and $l$ is the class the model misclassified examples from the class $k$ to. We show example pairs from different confusion types defined in Section 5.

In Figure 7 we show average original and ReaL accuracy on the 104 classes which have less than 1300 examples in the ImageNet train split (while the remaining majority of classes have exactly 1300 examples in train data). We hypothesize that if one of the confused classes has less examples in the train data and DA leads to higher class distribution overlap, the underrepresented class will likely be affected in accuracy by stronger augmentation.

# E Additional details for the class-conditional augmentation intervention experiments

In Figures 8 and 9 we show how the number of False Positive (FP) mistakes changes with data augmentation strength for the set of classes where FP number increased the most with strong DA (see Figure 8 for the set of classes where original FP mistakes increased the most and Figure 9 for ReaL FP mistakes). In Section 6, we conducted class-conditional data augmentation interventions changing the DA strength for these sets of classes and showed that it improved the accuracy on the classes negatively affected in accuracy. While in Section 6 we show results for adapting augmentation level for classes using original labels to evaluate False Positive and False Negative mistakes, in Table 3 we show analogous results when using ReaL labels which also shows that this targeted augmentation policy intervention for a small number of classes leads to improvement in ReaL average accuracy on the affected classes (we specifically consider the set of classes affected in ReaL accuracy).

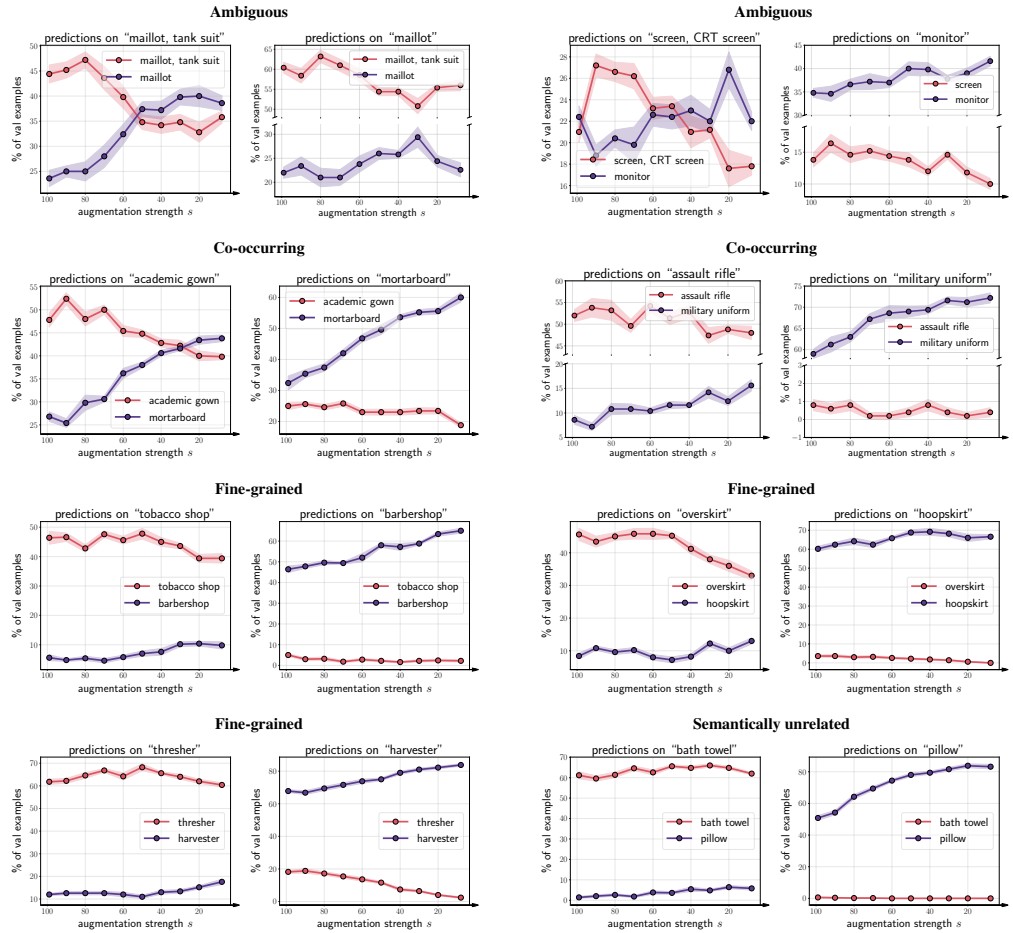

Figure 6: Confusion rate for classes most negatively affected by strong data augmentation and the corresponding classes they get confused with. We categorize confusions into ambiguous, co-occurring, fine-grained and unrelated.

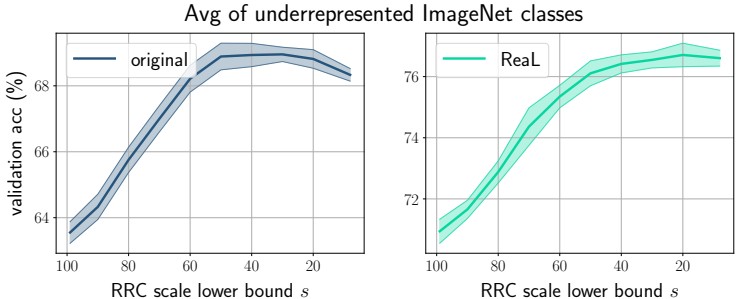

Figure 7: Average accuracy of the 104 underrepresented ImageNet classes depending on the Random Resized Crop augmentation strength. Average original accuracy on the underrepresented subset of ImageNet decreases by 0.6% with stronger augmentation.

Table 2: Confusions on the classes most affected by data augmentation.

| Affected class $k$ | Confused class $l$ | $\Delta$ conf. rate (%) | | Label co-occur. | | Semantic sim. | | Confusion type |
|---|---|---|---|---|---|---|---|---|
| | | $\Delta CR_{k\to l}$ | $\Delta CR^*_{l\to k}$ | $C_{lk}$ | IoU | WN | spacy | |
| overskirt | hoopskirt | 5.80 | 3.60 | 0.31 | 0.17 | 0.91 | – | fine-gr. (ambig.) |
| | bonnet | 4.20 | 0.00 | 0.03 | 0.02 | 0.73 | 0.32 | fine-gr. |
| | gown | 4.00 | 2.40 | 0.50 | 0.21 | 0.73 | 0.37 | fine-gr. (ambig.) |
| | trench coat | 3.60 | 0.40 | 0.00 | 0.00 | 0.75 | 0.42 | fine-gr. |
| academic gown | mortarboard | 18.40 | 7.00 | 0.72 | 0.50 | 0.73 | 0.10 | co-occur. |
| sunglass | sunglasses | 13.00 | 22.40 | 0.87 | 0.81 | 0.64 | 0.84 | ambig. |
| maillot | maillot | 15.00 | 7.20 | 0.73 | 0.63 | 0.70 | 1.00 | ambig. |
| Windsor tie | suit | 7.20 | 4.00 | 0.61 | 0.32 | 0.82 | 0.24 | co-occur. |
| screen | desktop computer | 7.80 | 7.00 | 0.59 | 0.29 | 0.64 | 0.62 | ambig. |
| | monitor | 3.20 | 6.40 | 0.87 | 0.37 | 0.63 | 0.44 | ambig. |
| tobacco shop | barbershop | 5.20 | 2.80 | 0.00 | 0.00 | 0.91 | 0.56 | fine-gr. |
| | bookshop | 6.80 | 6.40 | 0.00 | 0.00 | 0.91 | 0.53 | fine-gr. |
| monastery | church | 2.80 | 6.80 | 0.11 | 0.03 | 0.70 | 0.71 | fine-gr. |
| | castle | 2.80 | 11.20 | 0.00 | 0.00 | 0.60 | 0.69 | fine-gr. |
| thresher | harvester | 6.60 | 16.40 | 0.04 | 0.01 | 0.90 | 0.49 | fine-gr. |
| parallel bars | horizontal bar | 3.20 | 2.80 | 0.00 | 0.00 | 0.90 | 0.75 | fine-gr. |
| | balance beam | 3.00 | 4.00 | 0.02 | 0.01 | 0.90 | 0.45 | fine-gr. |
| mailbag | purse | 12.80 | 2.00 | 0.10 | 0.06 | 0.89 | 0.19 | fine-gr. |
| | backpack | 4.00 | 5.60 | 0.00 | 0.00 | 0.89 | 0.16 | fine-gr. |
| chain | necklace | 9.40 | 4.40 | 0.15 | 0.09 | 0.53 | 0.31 | ambig. |
| bulletproof vest | military uniform | 5.60 | 3.40 | 0.31 | 0.13 | 0.76 | 0.38 | co-occur. (ambig.) |
| | assault rifle | 3.20 | 0.40 | 0.32 | 0.17 | 0.40 | 0.35 | co-occur. |
| sombrero | cowboy hat | 7.40 | 4.80 | 0.15 | 0.05 | 0.91 | 0.51 | fine-gr. |
| velvet | purse | 3.60 | 2.60 | 0.00 | 0.00 | 0.62 | 0.29 | unrelated |
| | necklace | 3.00 | 0.00 | 0.00 | 0.00 | 0.62 | 0.51 | unrelated |
| tape player | radio | 3.20 | 4.60 | 0.00 | 0.00 | 0.67 | 0.27 | fine-gr. |
| | cassette player | 3.00 | 0.20 | 0.08 | 0.01 | 0.89 | 0.85 | fine-gr. |
| assault rifle | military uniform | 8.40 | 0.40 | 0.47 | 0.24 | 0.42 | 0.42 | co-occur. |
| cornet | trombone | 4.80 | 2.40 | 0.23 | 0.14 | 0.91 | 0.41 | fine-gr. |
| pole | traffic light | 4.00 | 0.40 | 0.05 | 0.03 | 0.12 | 0.21 | unrelated |
| muzzle | sandal | 3.20 | 0.00 | 0.00 | 0.00 | 0.56 | 0.23 | unrelated |
| ear | corn | 5.40 | 4.40 | 0.81 | 0.52 | 0.78 | 0.23 | ambig. |
| vault | altar | 6.40 | 4.40 | 0.21 | 0.12 | 0.62 | 0.41 | fine-gr. (ambig.) |
| frying pan | Dutch oven | 6.00 | 3.00 | 0.00 | 0.00 | 0.40 | 0.59 | fine-gr. |
| | wok | 3.40 | 2.60 | 0.09 | 0.05 | 0.92 | 0.72 | fine-gr. |
| French loaf | bakery | 4.40 | 1.80 | 0.10 | 0.06 | 0.24 | 0.42 | co-occur. |
| barrel | rain barrel | 7.60 | 2.20 | 0.16 | 0.07 | 0.76 | 0.70 | fine-gr. (ambig.) |
| spatula | wooden spoon | 4.40 | 2.80 | 0.24 | 0.12 | 0.57 | 0.62 | fine-gr. |
| sax | flute | 3.20 | 0.40 | 0.00 | 0.00 | 0.83 | 0.65 | fine-gr. |
| seashore | sandbar | 3.80 | 2.80 | 0.64 | 0.47 | 0.57 | 0.69 | co-occur. |
| coffee mug | cup | 7.80 | 0.80 | 0.61 | 0.34 | 0.19 | 0.63 | ambig. |
| | espresso | 3.00 | 2.60 | 0.18 | 0.13 | 0.21 | 0.72 | co-occur. |
| breastplate | cuirass | 6.00 | 6.40 | 0.71 | 0.50 | 0.67 | 0.48 | ambig. |
| | shield | 3.20 | 1.20 | 0.07 | 0.05 | 0.70 | 0.59 | |
| beacon | breakwater | 7.80 | 0.60 | 0.07 | 0.04 | 0.71 | 0.33 | co-occur. |
| suit | miniskirt | 3.20 | 1.60 | 0.02 | 0.01 | 0.86 | 0.32 | fine-gr. |
| hand-held computer | cellular telephone | 8.80 | 5.60 | 0.22 | 0.06 | 0.50 | 0.42 | ambig. |
| | notebook | 4.60 | 0.40 | 0.03 | 0.01 | 0.92 | 0.32 | fine-gr. |
| stopwatch | digital watch | 4.80 | 0.60 | 0.00 | 0.00 | 0.83 | 0.62 | fine-gr. |
| strawberry | trifle | 4.40 | 1.40 | 0.06 | 0.03 | 0.32 | 0.40 | co-occur. |
| trimaran | catamaran | 4.80 | 1.40 | 0.18 | 0.09 | 0.92 | 0.60 | fine-gr. |
| digital clock | digital watch | 3.00 | 7.00 | 0.02 | 0.01 | 0.83 | 0.71 | fine-gr. |
| hair slide | necklace | 5.60 | 0.60 | 0.00 | 0.00 | 0.50 | 0.42 | fine-gr. |
| hook | necklace | 3.60 | 0.00 | 0.00 | 0.00 | 0.53 | 0.33 | unrelated |
| backpack | purse | 3.00 | 0.00 | 0.02 | 0.01 | 0.89 | 0.56 | fine-gr. |
| home theater | monitor | 2.80 | 0.00 | 0.03 | 0.00 | 0.56 | 0.18 | co-occur. |
| bath towel | pillow | 4.40 | 0.60 | 0.00 | 0.00 | 0.59 | 0.56 | unrelated |

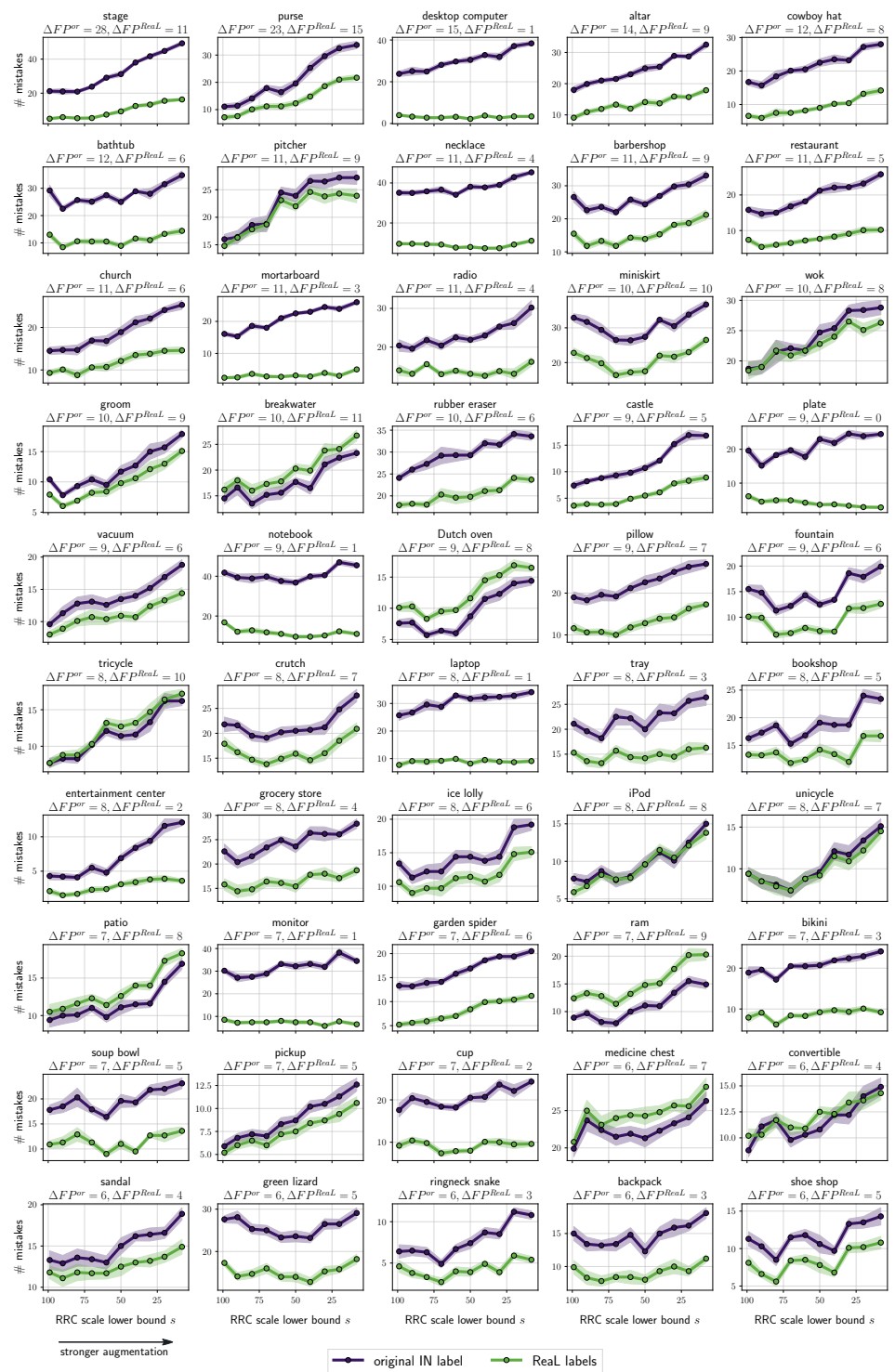

Figure 8: The number of per-class False Positive (FP) mistakes for the set of classes where FP computed with original labels increases the most when using strong data augmentation. We show the trends using both original and ReaL labels.

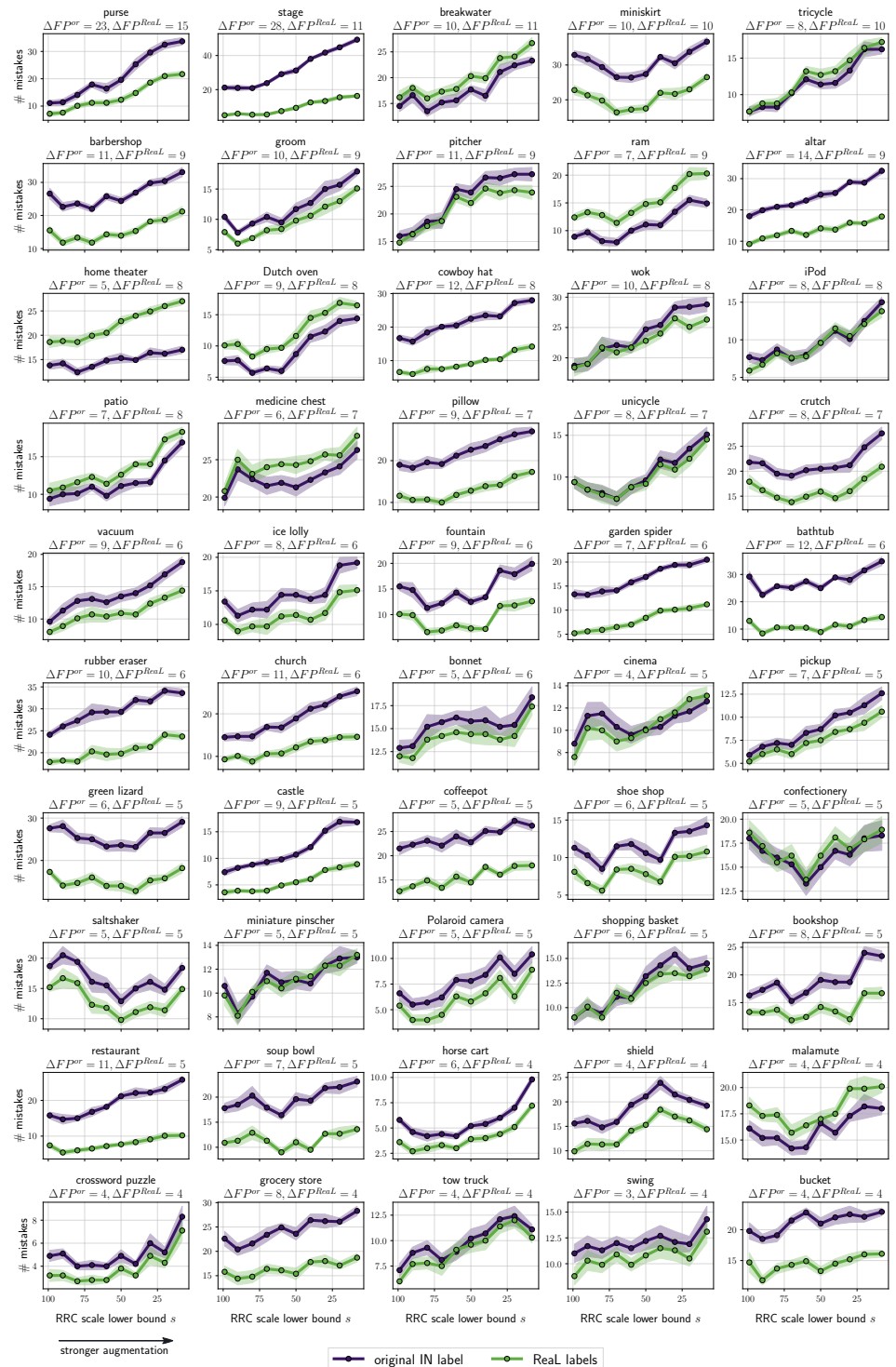

Figure 9: The number of per-class False Positive (FP) mistakes for the set of classes where FP computed with ReaL labels increases the most when using strong data augmentation. We show the trends using both original and ReaL labels.

Table 3: Class-conditional augmentation intervention using ReaL labels.

| # classes with adapted aug. | ReaL avg acc | ReaL avg acc of 50 aff. classes | ReaL avg acc of remaining 950 classes |
|---|---|---|---|
| $m = 0$ | $83.70_{\pm 0.01}$ | $70.66_{\pm 0.08}$ | $84.00_{\pm 0.01}$ |
| $m = 10$ | $83.63_{\pm 0.01}$ | $72.01_{\pm 0.04}$ | $83.86_{\pm 0.01}$ |
| $m = 30$ | $83.64_{\pm 0.01}$ | $72.28_{\pm 0.05}$ | $83.86_{\pm 0.01}$ |
| $m = 50$ | $83.57_{\pm 0.01}$ | $72.20_{\pm 0.03}$ | $83.78_{\pm 0.01}$ |

We also experimented with fine-tuning the model from the checkpoint trained with the strongest augmentation $s = 8\%$ using either regular augmentation policy which was used during training or class-conditional policy with augmentation strength changed for $k = 10$ classes as in Section 6: we fine-tuned the model for 5 epochs with linearly decaying learning rate starting from the value $10^{-4}$. However, both regular and class-conditional DA lead to slight drop in average accuracy on all classes (from 76.79% to 76.73% for either DA) and in particular the accuracy dropped more significantly for negatively affected classes: from 53.93% to 53.4%. We hypothesize that this is due to model memorizing train examples so even class-conditional augmentation policy is not able to recover performance on the affected classes if we re-use the same data for fine-tuning. In the future analysis, we will explore whether it is possible to alleviate DA bias if we fine-tune the model from an earlier checkpoint as opposed to fully trained model or if we use additional held-out data for fine-tuning.

# F  Additional architecture results: EfficientNet and ViT

In Figures 10 and 11 we show the per-class accuracy trends for classes most affected in original and ReaL accuracy of EfficientNet-B0 [61] model, trained using a similar setup to the main ResNet-50 model (see Section A). We can see that many affected classes are the same for ResNet-50 and EfficientNet models.

We also train a Vision Transformer model ViT-S [58, 65] varying the RRC scale lower bound in the range $s \in \{10\%, 20\%, \dots, 90\%\}$ and report the results in Figure 12. Generally, we confirm that our observations hold for ViT. While the optimal average accuracy is obtained with the strongest augmentation, for several classes accuracy significantly degrades. Evaluation with multi-label annotations reveals that some of the confusions are due to inherent label ambiguity or class overlap. We also identify the same high-level class confusion categorized as ambiguous, co-occurring, fine-grained and semantically unrelated (see Fig 13). By conducting a data augmentation intervention from Section 6 of the paper and changing the RRC augmentation strength for just 10 classes, we improve the accuracy on the degraded classes by over 3% (from 52.28% ± 0.18% to 55.49% ± 0.07%).

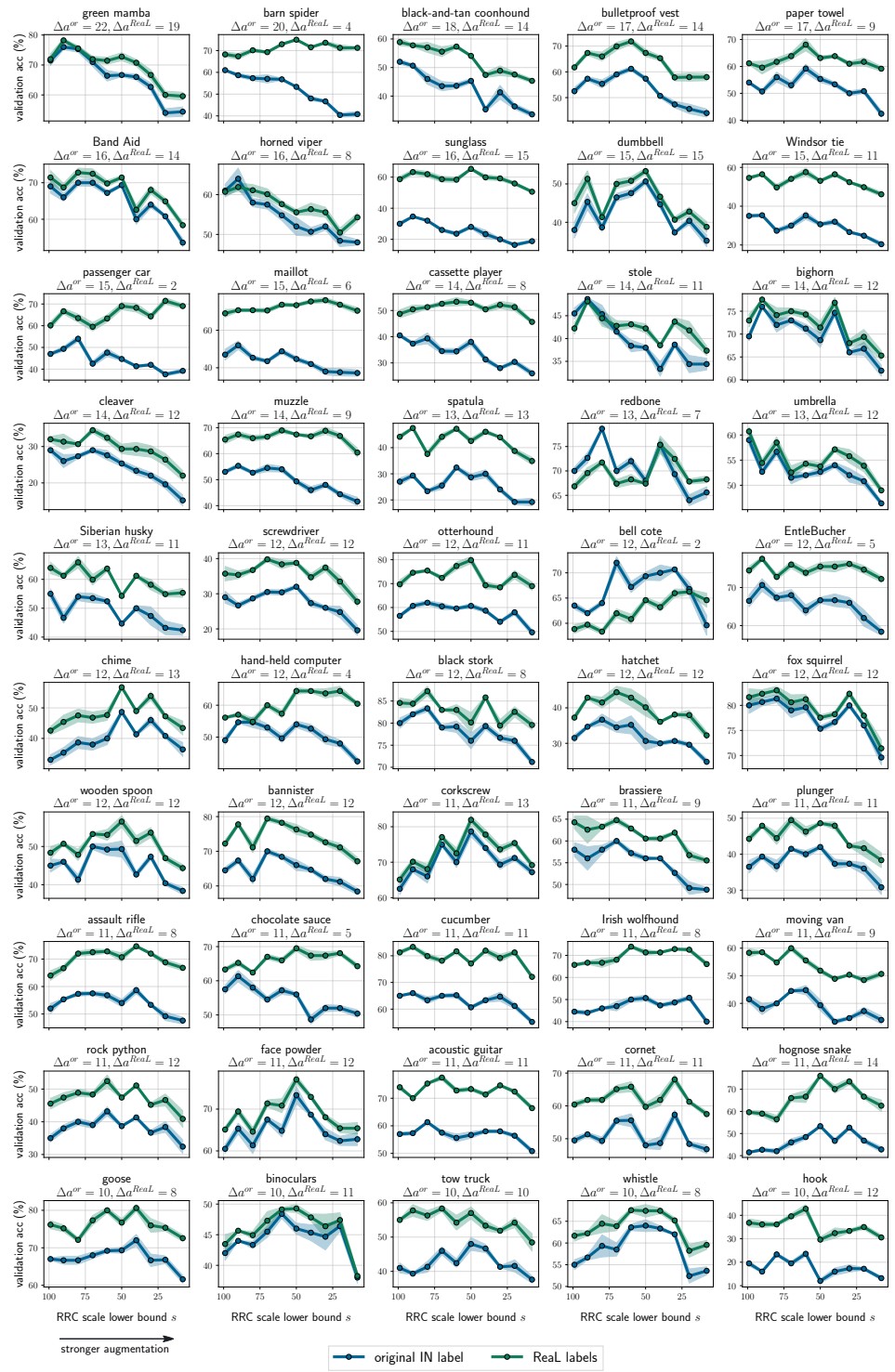

Figure 10: Per-class class validation accuracies of EfficientNet-B0 trained on ImageNet computed with original and ReaL labels as a function of Random Resized Crop data augmentation scale lower bound $s$. We show the accuracy trends for the classes with the highest difference between the maximum accuracy on that class across augmentation levels $\max_s a_k^{or}(s)$ and the accuracy of the model trained with $s = 8\%$. On each subplot below the name of the class we show the accuracy drops with respect to original and ReaL labels: $\Delta a_k^{or}$ and $\Delta a_k^{ReaL}$.

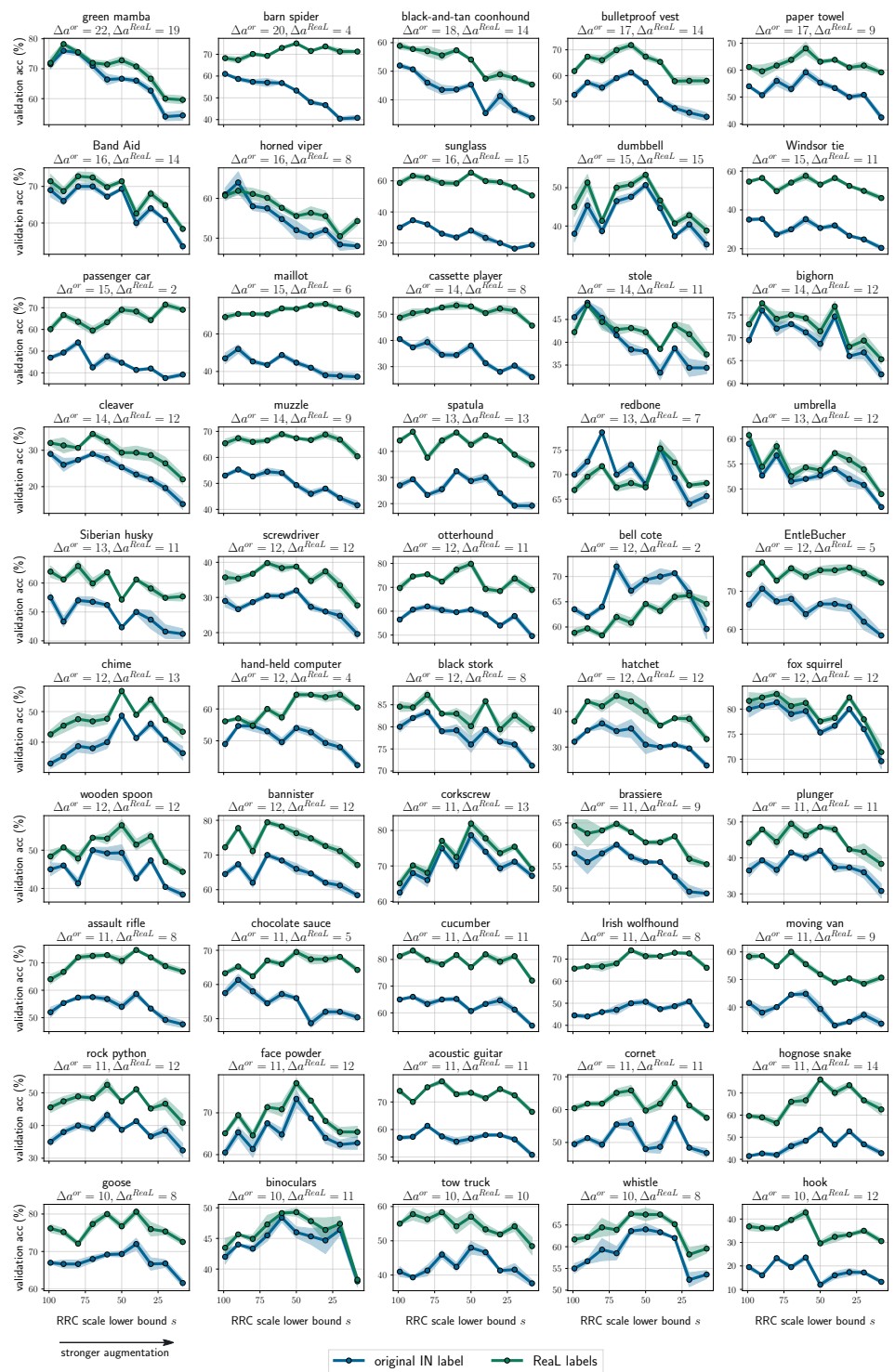

Figure 11: Per-class class validation accuracies of EfficientNet-B0 trained on ImageNet computed with original and ReaL labels as a function of Random Resized Crop data augmentation scale lower bound $s$. We show the accuracy trends for the classes with the highest difference between the maximum ReaL accuracy on that class across augmentation levels $\max_s a_k^{or}(s)$ and the ReaL accuracy of the model trained with $s = 8\%$. On each subplot below the name of the class we show the accuracy drops with respect to original and ReaL labels: $\Delta a_k^{or}$ and $\Delta a_k^{ReaL}$.

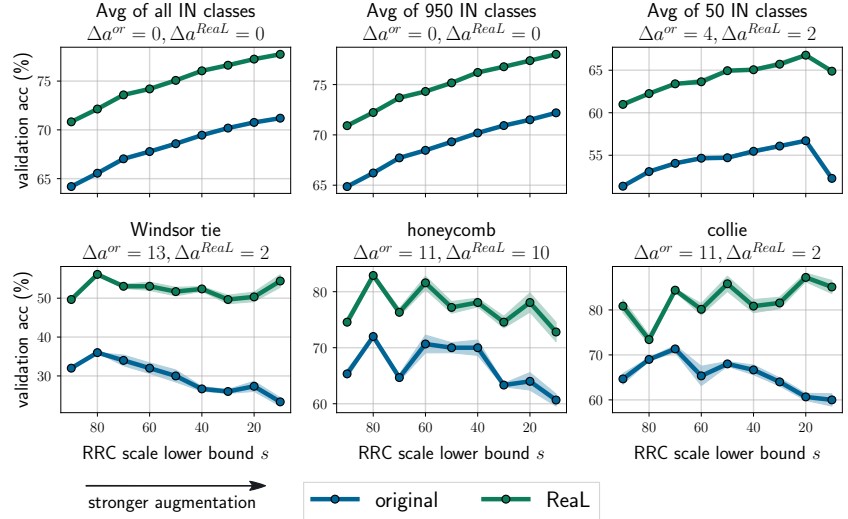

Figure 12: **ViT-S trained on ImageNet with varied Random Resized Crop (RRC) augmentation strength.** Average and per-class accuracy of ViT-S evaluated with original and ReaL labels as a function of RRC augmentation strength. The top row shows the average accuracy of all classes, the 50 classes with the highest accuracy degradation and the remaining 950 classes. The bottom row shows the accuracy of 3 classes most significantly affected in accuracy when using strong augmentation.

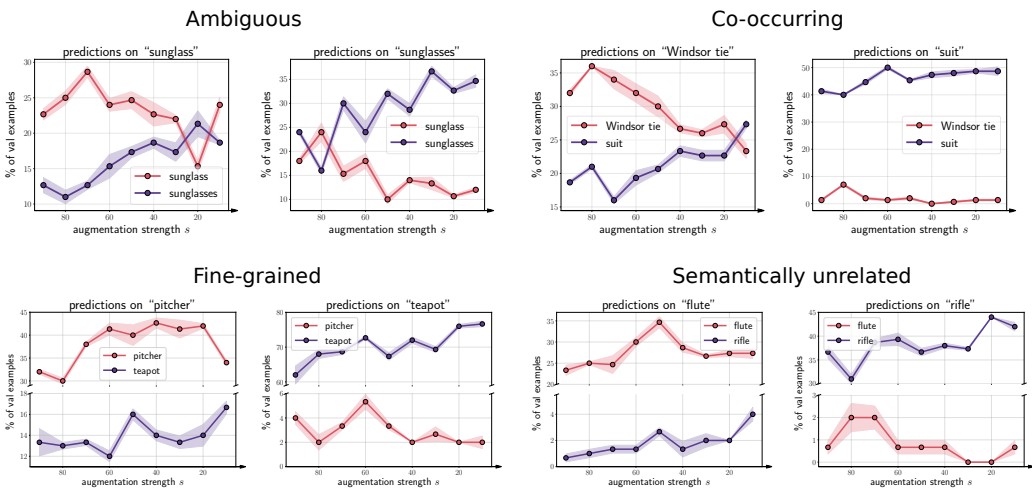

Figure 13: **Class confusion types of ViT-S model trained on ImageNet with varied Random Resized Crop (RRC) augmentation strength.** Each panel shows a pair of confused classes which we categorize into: *ambiguous*, *co-occurring*, *fine-grained* and *semantically unrelated*. For each confused class pair, the left subplot corresponds to the class $k$ affected in accuracy by strong data augmentation (DA), e.g. "sunglass" on top left panel: the ratio of validation samples from that class that get classified as $k$ decreases with stronger DA, while the confusion rate with another class $l$ (e.g. class "sunglasses" on top left panel) increases. The right subplot shows the percent of examples from class $l$ that get classified as $k$ or $l$ against DA strength. Generally, we observe that for ViT-S the class confusions which are exacerbated with stronger DA are similar to the ones of ResNet-50.

# G  Additional augmentation transformations and datasets

## G.1  Additional augmentation transformations

**RandAugment**    RandAugment [13] randomly applies color perturbations, translations and affine transformations. We train ViT-S with the RRC $s = 10\%$ and vary the RandAugnment magnitude $m$ in the range $\{1, 3, \ldots 9\}$ ($m = 9$ is standard for ViT [58, 65]; values above $m = 9$ lead to significant degradation in average accuracy). We report results in Fig 14. While RandAugment strength has a smaller effect on accuracy than RRC, we still observe an increase of around $0.5\%$ in average performance with $m = 9$. However, that comes at the cost of about $4\%$ accuracy drop for a minority of classes. In Figure 14 we show examples of class confusions exacerbated by RandAugment, which we can categorize analogously to Section 5.

**Colorjitter**    We train a ResNet-50 using the strongest RRC augmentation with colorjitter applied with probability $0.5$ and intensity $c = 0.1$ for all parameters (brightness, contrast, saturation and hue). Applying colorjitter with higher probability or intensity leads to degraded average accuracy while colorjitter leads to a slight improvement of $0.1\%$ in average accuracy. In Figure 15 we show the distribution of accuracy improvements and degradations due to colorjitter, as well as the examples of class confusions that were exacerbated.

## G.2  Additional augmentation transformations and datasets

**CIFAR-100 + mixup**    We study mixup [73] augmentation for ResNet18 on CIFAR-100. We train for 100 epochs using Random Crop, and mixup with $\alpha = 0.5$ improves the average accuracy from $78.11 \pm 0.15\%$ to $78.53 \pm 0.35\%$ However, we observe degradations for some per-class accuracies (see Figure 16). The exacerbated confusions are mainly within the same superclass categories of CIFAR-100 which is aligned with our prior results on ImageNet where we observed that fine-grained confusions are more significantly affected by augmentation.

**Flowers102**    We study the effect of applying standard RRC in Flowers102 classification task, and while using augmentation improves the average accuracy by $2\%$, we observe that some classes are negatively affected (see Figure 16).

# H  Broader impact and limitations

In our analysis we mainly focused on the setup of analyzing class-level accuracy drops of ResNet-50 model trained on ImageNet with varied Random Resized Crop (RRC) data augmentation. In Sections F and G we confirm our observations on additional architectures (EfficientNet-B0 and ViT-S), data

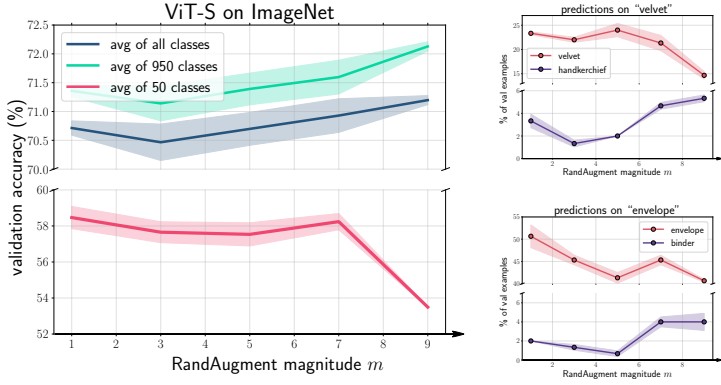

Figure 14: **Different augmentation types: RandAugment.** ViT-S model trained on ImageNet with varied RandAugment magnitude $m$ (larger values of $m$ correspond to stronger augmentation). The left panel shows the average accuracy of all ImageNet classes, the 50 classes with the highest accuracy degradation and the remaining 950 classes. The right panels show the examples of class confusions which are exacerbated by stronger RandAugment augmentation.

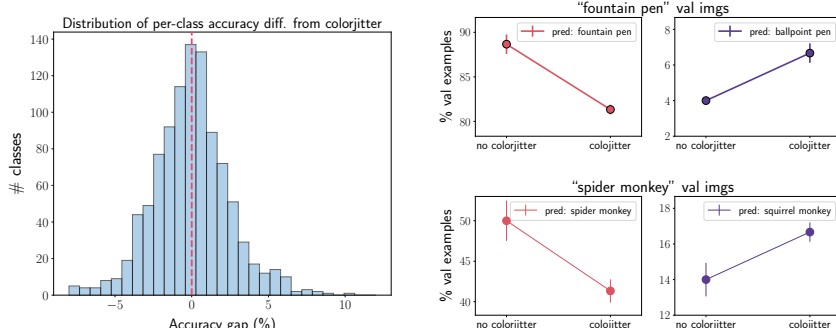

Figure 15: **Different augmentation type: colorjitter.** Comparison of ResNet-50 trained on ImageNet with Random Resized Crop $s = 8\%$ when using and not using colorjitter augmentation (applied with probability $p = 0.5$ and intensity $c = 0.1$). The histogram shows per-class accuracy changes when applying versus not applying colorjiter: most classes benefit from augmentation, while a significant number of classes is negatively affected. The right panels show the examples of class confusions exacerbated by applying colorjitter.

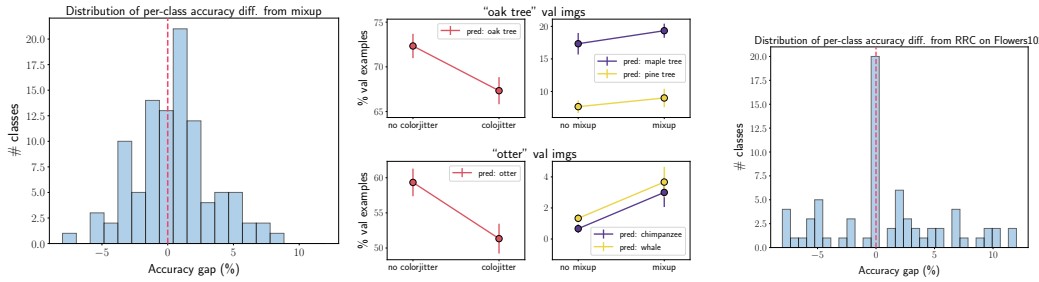

Figure 16: **Different datasets: CIFAR-100 and Flowers102. Left**: The histogram shows per-class accuracy changes when applying versus not applying mixup with $\alpha = 0.5$ when training on CIFAR-100. **Middle**: The examples of class confusions exacerbated by applying mixup on CIFAR-100, the exacerbated confusions are mostly fine-grained and lie within CIFAR-100 super-classes. **Right**: The histogram shows per-class accuracy changes when applying versus not applying standard Random Resized Crop $s = 8\%$ when training ResNet-32 on Flowers102 dataset.

augmentation transformations (RandAugment, mixup and colorjitter) and datasets (CIFAR-100 and Flowers102). The same proposed framework can be extended to better understand the biases of other augmentations, architectures and satasets in the future work. While we provide quantitative metrics to describe each confusion type affected by data augmentation, the categorization is not strict due to the remaining noise in ReaL labels and imprecise word similarity metrics.

A potential negative outcome that can result from misinterpretation of our analysis in Section 4 is if the practitioners assume that data augmentation does not have any negative effects since we discover that previously reported performance drops were overestimated due to label noise. We emphasize that while some of the class-level accuracy drops were indeed due to label ambiguity or co-occurring objects, data augmentation does exacerbate model's bias and introduces class confusions (often between fine-grained categories but sometimes even for semantically unrelated classes that share visually similar features). We encourage researchers to carefully study the negative impact of DA using fine-grained metrics beyond average accuracy (such as per-class accuracy, False Positive mistakes and class confusions) to better understand its biases.

**Compute.** We estimate the total compute used in the process of working on this paper at roughly 5000 GPU hours. The compute usage is dominated by training models for different augmentation strengths (Section 4). The experiments were run on GPU clusters on Nvidia Tesla V100, Titan RTX, RTX8000, 3080 and 1080Ti GPUs.

