# Understanding the Detrimental Class-level Effects of Data Augmentation: Supplementary Material

## A Training details

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

## G  Multi-label annotations

In this work we use ReaL labels released in Beyer et al. [4] to account for the label noise in evaluation of per-class accuracy effects of data augmentation. A more recent work Vasudevan et al. [64] released re-assessed multi-label annotations for a half of the ImageNet validation set. Since they did not release the annotations for the entire validation set, we decided to use older and more commonly used ReaL labels. However, one could merge the two multi-label annotation sets from Beyer et al. [4] and Vasudevan et al. [64] for more accurate evaluation. In particular, Vasudevan et al. [64] discussed the class mappings that they collapsed, and among those classes are the ones negatively affected in ReaL accuracy by data augmentation, e.g. "siberian huskies are also eskimo dogs", "coffee mug is also a cup", "maillot and maillot, tanksuit are the same class" "monitor and screen are the same class", "cassette player is also a tape player" [64].

## H  Broader impact and limitations

**Limitations.**   In this paper we consider the impact of Random Resized Crop (RRC) data augmentation which is the most commonly used augmentation transformation which is also often used in combination with other automatic augmentation policies [12, 42]. RRC DA also leads to most substantial improvements in average accuracy, unlike other transformations such as color-based augmentation which usually leads to limited improvements. For the main analysis we focus on ResNet-50 architecture and study per-class accuracies of EfficientNet-B0 [57] in Section I, however, Balestriero et al. [1] showed that per-class biases persist in other architectures like Vision Transformers [14] and DenseNets [29] and for colorjitter augmentation. While we provide a deep analysis of RRC per-class effects in ResNet models, the same