# OpenReview forum: "Understanding the detrimental class-level effects of data augmentation"
_NeurIPS.cc/2023/Conference — NeurIPS 2023 poster_

### Official Review · Reviewer_wB8d · 2023-06-25

**Soundness:** 3 good
**Presentation:** 3 good
**Contribution:** 3 good
**Rating:** 6
**Confidence:** 4

**Summary:**

This paper studies Data augmentation (DA). Although DA improves overall accuracy, recent studies have pointed out that it can adversely affect individual class accuracy by up to 20% on ImageNet. This happens due to a lack of understanding of how DA impacts class-level learning dynamics. This research offers a framework to understand this interaction. By using high-quality multi-label annotations on ImageNet, it is found that most affected classes are inherently ambiguous, co-occurring, or involve fine-grained distinctions. Although multi-label annotations explain many previously reported performance drops, the analysis reveals other sources of accuracy degradation. It is demonstrated that class-conditional augmentation strategies, informed by this framework, can enhance the performance on classes negatively impacted by DA.


--------Post-rebuttal------------

The rebuttal has fairly addressed my concerns. Accordingly, I have improved my score.

**Strengths:**

This study conducts an analysis of data augmentation bias, a process that bears significant practical implications for real-world applications. The meticulous examination of such bias provides insight into how it affects and shapes models in practical scenarios.

The research specifically investigates the impact of random crop scale on model training. It is revealed that the class-level performance drop experienced during ImageNet training can be mitigated through multi-label annotation. Intriguingly, the primary cause of this drop is found to be the co-occurring, ambiguous, and noisy conditions present within the class labels. This discovery is not only fascinating but potentially inspiring, opening up new avenues for further exploration in this domain.

The study introduces a novel class-conditional data augmentation strategy. This innovative approach shows promise in further alleviating data augmentation bias, thereby enhancing the robustness and performance of models in real-world deployments.

**Weaknesses:**

This work is limited by only one augmentation method (i.e., random crop ) and only CNN-based architectures (ResNet and EfficientNet).

**Questions:**

It would be beneficial to extend the analysis to encompass a wider range of augmentation methods, such as the hyperparameters in RandAugment and Data MixUp. This could potentially ascertain whether the observations drawn from the study are sufficiently generalized across various data augmentation methods, thus enhancing the validity and universality of the findings.

Are the observations consistent when applied to Vision Transformer (ViT)-based models? Although this study demonstrates the consistency of findings between ResNet and EfficientNet (both are Convolutional Neural Network (CNN) architectures), providing additional analysis with ViT-based models could significantly strengthen the overall solidity of the paper. This extended investigation would provide a more comprehensive understanding of how different model architectures interact with the described phenomenon.

**Limitations:**

Please refer to the weaknesses and questions.

---

> ### Author Rebuttal · Authors · 2023-08-10
>
> We thank the reviewer for their clear understanding of our submission and for noting the importance of our study! Please see our general response in which we detail our findings on new datasets, architectures and data augmentations (notably including the suggested ViT model). We observe that our insights, methodology, and DA intervention transfer to those different settings. As a brief summary, added experiments include one non-convolutional architecture (ViT, as per your suggestion), and three additional DAs (RandAugment, mixup, and colorjitter), and two new image datasets (CIFAR100 and Flowers102). We hope you can consider our responses in your final evaluation.

---

> > ### Comment · Reviewer_wB8d · 2023-08-12
> >
> > Thanks for the response. I believe, with these new experimental results, this paper will become more solid and thorough.
> >
> > I have another question driven by curiosity. I believe the following analysis could be insightful: could you provide a deeper analysis or make a summary on why label noise and label co-occurring result in performance drops and false positives with strong augmentation? I'm not asking for additional experiments, just curious about the essential reasons.

---

> > > ### Author Response · Authors · 2023-08-14
> > > **Thank you for your response**
> > >
> > > Thank you for engaging in further discussion, and for your review feedback which helped us expand the scope of our work and made the submission stronger with the new analysis and experimental results!
> > >
> > > In our paper we showed that among the classes that are negatively affected by strong data augmentation are often ambiguous, co-occurring and fine-grained categories (measured by co-occurrence frequency in ReaL labels and semantic similarity of categories).
> > > To explain the intuition behind the class-level accuracy drops and false positive mistake increase, we can take co-occurring classes “academic gown” and “mortarboard” as an example, however, similar mechanisms apply to other confusion types as well. Since 60% of the examples with label “mortarboard” also have ReaL label “academic gown”, sometimes  random resized crops (RRC) of the images labeled as “mortarboard” will focus on the gown but the model would still be optimized to predict the “mortarboard” label on such training examples. As a result, the model will become biased to predict “mortarboard” on “academic gown” images, leading to degradation in “academic gown” accuracy, and increased false positive mistakes for “mortarboard”. In general, we will observe similar effects whenever strong RRC applied to class $l$ will often focus on the features that effectively correspond to the features of another class $k$: this will lead to degradation in accuracy for class $k$ and increase in false positive mistakes for class $l$.
> > >
> > > For ambiguous or nearly identical categories, such as “sunglass” and “sunglasses”, the classes might have slight differences induced by idiosyncrasies of the labeling pipeline (such as the images from one of the classes are more often zoomed in than from the other class), and due to such statistical differences the data augmentation strength will control the bias towards one of the two plausible categories.
> > >
> > > Please let us know whether this answers your question, we will be happy to further provide clarifications or answer any other questions during the discussion period!

---

> > > > ### Comment · Reviewer_wB8d · 2023-08-15
> > > >
> > > > Great! I do not have further concerns. Accordingly, I have improved my score. Good luck.

---

### Official Review · Reviewer_EjrR · 2023-07-03

**Soundness:** 4 excellent
**Presentation:** 4 excellent
**Contribution:** 3 good
**Rating:** 7
**Confidence:** 3

**Summary:**

The authors explore the role of random resize crop in ImageNet performance. First, they improve on analysis in prior work and show that class-level performance degradation has been over-stated, and that when multi-label annotations are used one of the labels is often still predicted. Next, by inspection, the authors find that many  classes whose accuracies are affected random-resize crop are either completely ambiguous (sunglass vs sunglasses) or co-occurring (suit, tie; car, wheel). Finally, they suggest an intervention which takes false positives into account in addition to false negatives, dramatically improving accuracy on most-affected classes while preserving overall accuracy.

**Strengths:**

Originality: The idea of using multi-label annotations to judge the severity of the mistakes is sensible.
The idea of using False Positives for conditioning augmentation is an interesting one.
In general, the idea of using labels in the training procedure as a conditioning mechanism outside of the loss is interesting.

Quality: The methodology seems sound and the claims are supported by the evidence.

Clarity :The paper is clearly written and the figures are well understood.

Significance: The paper is significant in that it sheds light on a process which is often used as a black box (data augmentation), and moves us towards more bespoke models. It is the case that we often report best top-1 accuracies, averaged over a validation set, and that this obscures the variation in different kinds of mistakes. However, certain kinds of mistakes are much more expensive than other kinds. Then, improving performance for certain classes while retaining performance of others is an important intervention for developing deployable models.  This work takes a step towards the understanding needed for making these kinds of trade offs.

**Weaknesses:**

* The analysis is done only on a single model (supervised ResNet-50), on a single dataset (ImageNet). While I expect conclusions to be similar across other datasets/models, it would be interesting and important to confirm this.

* Similarly, only consider RRC. What about other common augmentations?

* ImageNet pretraining is frequently used for transfer learning. Does the class-conditional augmentation intervention impact transfer performance?

**Questions:**

See weaknesses.

* Does the intervention have more impact on learned features, or on the final classification layer?  In other words, if the backbone was frozen after standard training and the final layer was retrained using the proposed intervention, what would the effect be?

**Limitations:**

Authors have adequately addressed limitations.

---

> ### Author Rebuttal · Authors · 2023-08-10
>
> Thank you for your insightful review! We appreciate that you find our work interesting, significant and clearly written.
>
> **Model, dataset, data augmentation**: We agree that confirming the applicability of our insights and methods to other models and datasets is crucial. To that end, we have provided supporting experiments which include a non-convolutional architecture (ViT-S), two datasets (CIFAR100, Flowers102), and three DAs (colorjitter, mixup and RandAugment). While we describe in detail the experiments and findings in the general answer, we mention here that our conclusions hold in those settings, and that the intervention we propose also transfers to those cases. We hope that those findings will provide further evidence for the validity of our findings in the broader context of computer vision.
>
> **Effects on learned features vs final layer**.  We have explored a similar experiment in the Appendix F of our submission: we experimented with fine-tuning the model from the checkpoint trained with the strongest augmentation using class-conditional policy with augmentation strength changed for 10 classes as in Section 6. However, we found that such fine-tuning leads to a drop in average accuracy and accuracy of the negatively affected classes. We hypothesize that this might be due to a feature extractor learning unwanted invariances or model memorizing the training data. In the future analysis, we will explore whether it is possible to alleviate DA bias if we fine-tune the model from an earlier checkpoint as opposed to fully trained model or if we use additional held-out data for fine-tuning.
>
> **Transfer learning**: This is a very interesting suggestion! We prioritized looking at different architectures, datasets and DAs for our rebuttal, but we agree that it is an important avenue that we hope to explore in the future.

---

> > ### Comment · Reviewer_EjrR · 2023-08-10
> > **Acknowledgement**
> >
> > The authors have sufficiently answered my concerns in the rebuttal, and I have decided to keep my ratting of of 7(accept).

---

### Official Review · Reviewer_nQVM · 2023-07-05

**Soundness:** 2 fair
**Presentation:** 3 good
**Contribution:** 2 fair
**Rating:** 7
**Confidence:** 4

**Summary:**

The paper presents a meta study on the effects of data augmentation over classes. In particular, authors works on ResNet50 architectures trained on ImageNet, and show how for some classes, strong data augmentation drastically decreases the per-class accuracy. The paper focuses on random resized crop augmentations and specifically by varying the lowest possible crop size. Authors discuss these results in light of different types of overlap between classes and show how considering a different ground truth shows that the models are not always learning something wrong, but maybe learn to focus on object parts rather than whole objects.

**Strengths:**

- Interesting study overall, which extends prior work on the same topic. The paper exposes some effects that might be obvious, but does so in a systematic and well constructed manner.
- The paper is very well written, and well presented. It was easy to follow and to understand main points.
- Supplementary materials contain interesting plots and additional results nicely complementing the paper.

**Weaknesses:**

- Limited to one data augmentation strategy, it would be interesting to see how other potentially aggressive class-confusing augmentations effect the classification scores (eg. colour changes, contrast enhancements, affine transformations, etc.) interact with RRC and among them. I understand this would make such analysis considerably more complex, but in the end, those are commonly used and it rarely happens that one relies exclusively on RRC.
- The study is strongly limited by the use of ImageNet only, other datasets have multilabel annotations. It would be really beneficial to understand the effects on other datasets, where the total number of classes is lower and potentially the relatedness across classes also reduced.
- Similarly, it would be interesting to understand if other CNN models or ViT would behave the same. I think it would nicely close the circle. I think such study on different models and different datasets would be quite interesting overall.
- In conclusion, although well done and well presented, I think this paper does not make a substantial contribution in this current form, but could potentially do so if the breadth and scope are extended.
- I kept wondering what would happen if I train / fine tune the classifier in a multi-label setting, instead of just using ReaL as enhanced ground truth. It could be possible to do so using eg. coco?

**Questions:**

- L33: as pointed out above, it would be interesting to see the effects on other datasets where classes are less and ideally only a subset of classes interact. Is there any literature on this that can be used here?
- I would try to motivate why only the RRC is used. Be it simplicity, be it as an example of widely used augmentation, but I could not stop thinking in the paper how other DA strategies would behave.
- I wonder if some dataset annotations where object parts annotations are available (eg https://github.com/TACJu/PartImageNet) could not be used to study in more detail the relationship between local crop and object parts
- I missed at the end of Section 2 a summary of main limitations of current SOTA and to what this paper is answering specifically.
- I would try to motivate (section 3 beginning) why only focusing on RRC, ImageNet and ResNet50. I agree on the choices, but is there something more generic we can draw out of this? What if I replace the imageNet with a DenseNet or use ResNet to classify COCO in multilabel settings? I missed some more general guidelines and motivation on the study.
- L215-128 / L248-251: Would it be relevant to check if the model, in its top-K predictions, contains the classes that ReaL would show as most ambiguous? Independently of the object part relationship / co-occurrences / semantic similarity, a biased model should tend to predict those classes with high probabilities, maybe the predictions themselves and the co-occurrence of predictions could somehow show some of these effects.

**Limitations:**

The paper does not discuss limitations of the study, but I think some points mentioned above could be discussed, at least, to strengthen motivations and underline that some of the effects observed could be observed in general.

---

> ### Author Rebuttal · Authors · 2023-08-10
>
> Thank you for your feedback! Please also see our separate general post, which contains new experiments inspired by your comments and clarification on the setup. Inspired by your feedback we have significantly increased the scope of our paper, and included results for new architectures, datasets and data augmentation types. We hope you can consider our responses in your final evaluation, and please let us know if we can answer any additional questions during the discussion period.
>
> ### Limitations of previous studies and motivation and contributions of our work
>
> Although recent studies observed that data augmentation may lead to severe class-level accuracy drops [1, 2], no method has yet addressed this issue due to limited understanding on the origins of that performance degradation. To our knowledge, our work is the first systematic investigation narrowing down the root cause of class-level accuracy degradation when using DA. As a direct result of this finding, we propose a class-dependent DA intervention that takes into account the trade-off between class-level false negative and false positive mistakes. Unlike previous attempts to mitigate this issue [1], our data augmentation intervention solely changes the DA strength for a few classes, significantly improving the accuracy of the negatively affected classes, while retaining strong accuracy on average.
>
> ### The choice of the dataset, models and augmentation type, and additional experiments
>
> The focus on ImageNet stemmed from previous studies which reported class level accuracy drops primarily on that specific dataset[1, 2]. Inspired by your suggestion, we consider additional datasets to extend our insights beyond ImageNet, and include analysis on CIFAR-100 and Flowers102 in our general response.
>
> Similarly, our focus on RRC stems from it being the most impactful DA in terms of final average accuracy increase. Most computer vision models use RRC with [8%, 100%] bounds, sometimes combined with additional DAs. Yet, it is also natural to ask if our conclusions extend beyond RRC which is why we have considered RandAugment,  colorjitter, and mixup in our general answer.
>
> Beyond the dataset and DA, the choice of architecture is also crucial. While we explored ResNet and EfficientNets, we acknowledge that those belong to the same convolutional family, and thus, as per your suggestion, we have added ViT-S in our general answer.
>
> In all of these new experiments we found results to be consistent with our observations in the main paper. We hope that the addition of two datasets, one non-convolutional architecture, and three additional DAs help reinforce our methodology and conclusions.
>
> [1] Balestriero et al, The Effects of Regularization and Data Augmentation are Class Dependent, 2022
>
> [2] Bouchacourt et al, Grounding inductive biases in natural images:invariance stems from variations in data, 2021

---

> > ### Comment · Reviewer_nQVM · 2023-08-14
> > **Rebuttal Follow up**
> >
> > I would like to thanks Authors for taking the time and considerable effort to address my questions and limitations. The rebuttal and the additions to the paper are significant and indeed address the points I raised as limitations and weaknesses. The additional model (ViT-S) is very relevant, and complements well. Adding DA strategies also removes some levels of doubts regarding generality of the work. Additional analyses are relevant and informative.
> >
> > I might have been a bit strict on my previous review, I really wanted the scope of the work not to be limited by models and augmentations strategies. I like the separation of the modes of errors into visual ambiguity, class co-occurrence, fine grained distinctions, and semantically unrelated but likely visually interacting features. I think these aspects are important to be considered for a high-level understanding of what those models learn, but also how to best use them.
> >
> > After considering the rebuttal, the response to the review(s) and considering again the contribution, I think the research paper has now filled the gaps I felt were limiting the contribution, and I am happy to raise the score to Accept. I look forward to further studies relating how to use multi-label annotations and object parts to train more robust models, or have an idea how we could make data augmentation strategies be consistent with semantics and observers viewpoint (eg. object parts or full object class) to train better models.

---

> > > ### Author Response · Authors · 2023-08-14
> > > **Thank you for your response**
> > >
> > > Thank you very much for your feedback which helped us expand the scope of our work and for taking into consideration our response in your final score!

---

### Official Review · Reviewer_2Ged · 2023-07-06

**Soundness:** 3 good
**Presentation:** 3 good
**Contribution:** 3 good
**Rating:** 4
**Confidence:** 5

**Summary:**

The authors study the effect of data augmentations on the classwise performance under data imbalance. They focus on the rezised cropping operation and  distinguish between 4 different failure cases by using multi-label annotations. They also show that it is possible to recover some of them but using an informed class-conditional augmentation.

**Strengths:**

The tackled problem is very relevent since it aims to understand the failure cases introduced but the standard data augmentation pipeline.
The authors provide decent empirical support to their claims and the paper is overall well written.

**Weaknesses:**

* Although the introduced class conditional augmentation seems to help recovering some failures cases,  the overall performance of the model is not improving!  This unexpected behavior is worth more investigation

* The last category of failures denoted 'semantically unrelated' remains unexplained

* Minor: in line 267, 268: the authors claim that figure 3 shows the confusion between bath towel and pillow while it acctually shows the confusion between muzzle and sandal.

**Questions:**

Improving the qualit of this paper would definitely require adressing the major weakness cited above

---

> ### Author Rebuttal · Authors · 2023-08-10
>
> Thank you for your review and constructive feedback! We respond to your questions below, and we will remain available for any further discussion and clarifications throughout the discussion period.
>
> ### Average performance improvement
>
> The reviewer raises an interesting point regarding the average performance not improving upon adjusting the per-class DA strength. We emphasize that the classes on which our intervention changes the DA only account for a very small portion of the dataset (1-5%). As such, even in the ideal scenario in which we  significantly improve the accuracy on these classes, the impact on the average performance will be quite minimal. For example, a 10% accuracy improvement on 1% of the classes would only result in 0.1% improvement in average accuracy. Consequently, the goal of our intervention is to improve the performance specifically on the classes negatively affected by the default augmentation policy, and not the average performance. In the paper, we show that our intervention improves the performance on the target classes, while not significantly affecting the performance on the remaining classes, as expected.
>
> We also note that interventions targeted at improving performance disparities often significantly *hurt* the average performance, which is not the case with our intervention. Moreover, we believe that our observations could inspire future work, which would target improving the average performance, by designing more elaborate augmentation policies.
>
>
> ### Categories of class confusions
>
> One of our key conclusion is that DA (e.g. RRC) reduces the performance of the model on a minority of the classes by introducing confusing (image, label) pairs which could be attributed to different classes. The prototypical example we employed was on how the RRC of a car picture might produce a wheel picture. As such car and wheel can be considered as ambiguous, or related, under the RRC DA. Therefore, by monitoring the True Positive, False Positive, False Negative and True Negative rates of those two classes, a relation can be drawn (this is our main observation also enabling our per-class DA intervention).
>
> For the **semantically unrelated** confusion category, the classes often share similar visual features. For example, the classes muzzle and both show a net-like structure. Similarly, the classes flute and rifle share a long tube shape. Depending on the augmentation strength the model assigns objects with these features to one of the two conflicting classes.
>
> **Line 267, 268**: We thank the reviewer for their careful reading and for noticing that mistake, this will be corrected in our final submission.
>
> We hope the above answers have alleviated your concerns, but if you have any further questions we are happy to discuss at any point throughout the discussion period.

---

> > ### Comment · Area_Chair_wxUY · 2023-08-18
> > **Official comment by the area chair**
> >
> > Dear Reviewer,
> >
> > The author has posted their rebuttal, but you have not yet posted your response. Please post your thoughts after reading the rebuttal and other reviews as soon as possible. All reviewers are requested to post this after-rebuttal-response.

---

### Author Rebuttal · Authors · 2023-08-10

We thank all reviewers for their feedback! We are happy that the reviewers found our paper is “systematic and well constructed” (nQVM), “bears significant practical implications” (wB8d) and “sheds light on a process which is often used as a black box” (EjrR). Data augmentation (DA) is essential in deep learning, yet it is poorly understood. Recent studies observed that DA may lead to severe class-level accuracy drops [1, 2]. To our knowledge, our submission is the first systematic investigation of these degradations, narrowing down their cause and partially alleviating them. In particular:

(1) We correct prior work analysis on ImageNet using multi-label annotations: many previously reported cases of class-level performance degradation are explained by label ambiguity;

(2) We systematically categorize class confusions exacerbated by DA, using multi-label annotations and class similarities, revealing that the majority of these confusions concern ambiguous, spuriously correlated or fine-grained categories;

(3) We propose a class-dependent DA intervention inspired by our findings. Unlike previous attempts [1], our intervention only changes the DA strength for a few classes, and significantly improves the accuracy on the affected classes.

We believe that our work makes significant advances towards reliably deploying DA in the real world, where we must understand and remedy its detrimental effects to avoid costly mistakes. We have added multiple experiments inspired by reviewer feedback, and hope these results can be considered in the final assessment.

## Additional Experiments

The main feedback shared across reviewers was on the generality of our results beyond ImageNet, ResNet-50, and the Random Resized Crop (RRC). We have conducted several novel experiments during the rebuttal and significantly extended the scope of our work, as we describe below. We will add these results to our final submission.

### Vision Transformer

We train a Vision Transformer model ViT-S [3, 4] varying the RRC scale lower bound $s$ in the set {10%, 20%, …, 90%} and report the results in Figure 1 of the pdf. Generally, we confirm that our observations hold for ViT.

While the optimal average accuracy is obtained with the strongest augmentation, for several classes accuracy significantly degrades. Evaluation with multi-label annotations reveals that some of the confusions are due to inherent label ambiguity or class overlap (Figure 1 left). We also identify the same high-level class confusion categorized as ambiguous, co-occurring, fine-grained and semantically unrelated (Fig 1 right).

By conducting a data augmentation intervention from Section 6 of the paper and changing the RRC augmentation strength for just 10 classes, we improve the accuracy on the degraded classes by over 3% (from 52.28 $\pm$ 0.18% to 55.49 $\pm$ 0.07%).


### Additional data augmentation types

**RandAugment**: RandAugment [6] randomly applies color perturbations, translations and affine transformations. We train ViT-S with the RRC $s=10\%$ and vary the RandAugnment magnitude $m$ in the range {1, 3, … 9} ($m=9$ is standard for ViT [3, 4]; values above $m=9$ lead to significant degradation). We report results in Fig 2 left. While RandAugment strength has a smaller effect on accuracy than RRC, we still observe an increase of around 0.5% in average performance with $m=9$. However, that comes at the cost of about $4\%$ accuracy drop for a minority of classes. In Figure 2 we show examples of class confusions exacerbated by RandAugment, which we can categorize analogously to Section 5 of our paper.

**colorjitter**: We train a ResNet-50 using the strongest RRC augmentation with colorjitter applied with probability 0.5 and intensity $c=0.1$ for all parameters (brightness, contrast, saturation and hue). Applying colorjitter with higher probability or intensity leads to degraded average accuracy while colorjitter $c=0.1$ leads to a slight improvement of $0.1\%$ in average accuracy. In Figure 2 right we show the distribution of accuracy improvements and degradations due to colorjitter, as well as the examples of class confusions that were exacerbated.

### New datasets and augmentation

**CIFAR-100 + mixup**: We study mixup [7] augmentation for ResNet18 on CIFAR-100. We train for 100 epochs using Random Crop and mixup with $\alpha=0.5$ improves the average accuracy from $78.11 \pm 0.15\%$ to $78.53 \pm 0.35\%$. However, we observe degradations for some per-class accuracies (see Figure 3). The exacerbated confusions are mainly within the same superclass categories of CIFAR-100 which is aligned with our prior results on ImageNet where we observed that fine-grained confusions are more significantly affected by augmentation.

**Flowers102**: We study the effect of applying standard RRC in Flowers102 classification task, and while using augmentation improves the average accuracy by $2\%$, we observe that some classes are negatively affected (see Figure 3).

## Motivation of the setup

While we significantly increased the scope of our analysis in the rebuttal, we would like to clarify the motivation of the setup in our submission. We focus on ImageNet as it is a widely used large-scale benchmark, and the class-level accuracy drops were previously noticed specifically on ImageNet [1, 2]. We chose to study Random Resized Crop as it is used in training most modern state-of-the-art computer vision models [e.g. 3, 4, 5], and it leads to the most significant improvements in terms of average accuracy, compared to more modest accuracy gains achieved with colorjitter, mixup, or RandAugment.

### References

[1] https://arxiv.org/abs/2204.03632

[2] https://arxiv.org/abs/2106.05121

[3] https://arxiv.org/abs/2106.10270

[4] https://arxiv.org/abs/2012.12877

[5] https://arxiv.org/abs/2201.03545

[6] https://arxiv.org/abs/1909.13719

[7] https://arxiv.org/abs/1710.09412

---

### Decision · Program_Chairs · 2023-09-21

**Decision:**

Accept (poster)

**Comment:**

The submission studies the per-class impact of data augmentation on the ImageNet image classification benchmark. They focus on investigating CNN architectures and the random crop data augmentation. The authors identify some limitations of the prior studies on this topic (that the most affected classes are often inherently ambiguous, co-occurring, etc.), and demonstrate the effectiveness of class-conditional data augmentation strategies.

The submission received overall positive reviews, where the reviewers found the submission to study an interesting and significant problem,  the findings on the root causes of per-class performance degradation to be inspiring, and the paper to be well written. They also had concerns that the scope of experiments to be limited, studying only the CNN architecture with one type of data augmentation on one dataset. The authors provided a thorough rebuttal with additional studies on more model architecture (ViT), data augmentation types, and datasets. The rebuttal successfully addressed the concerns from three out of four reviewers, and they all recommended acceptance. Reviewer 2Ged did not respond to the rebuttal, and the AC has confirmed that the raised questions have been properly addressed by the rebuttal.

The AC thus recommends acceptance of the submission. The authors should make sure to incorporate the new results in the rebuttal into the final version.